# Post-Fishing Ban Period: The Fish Diversity and Community Structure in the Poyang Lake Basin, Jiangxi Province, China

**DOI:** 10.3390/ani15030433

**Published:** 2025-02-04

**Authors:** Chiping Kong, Yulan Luo, Qun Xu, Bao Zhang, Xiaoping Gao, Xianyong Wang, Zhen Luo, Zhengli Luo, Lekang Li, Xiaoling Gong

**Affiliations:** 1Jiujiang Academy of Agricultural Sciences, Jiujiang 332000, China; kongchiping@126.com (C.K.); 18138298841@163.com (Q.X.); 15339773958@163.com (B.Z.); ottocg131@126.com (X.G.); heartofocean9@163.com (X.W.); 16628729931@163.com (Z.L.); 2Key Laboratory of Exploration and Utilization of Aquatic Genetic Resources, Shanghai Ocean University, Ministry of Education, Shanghai 201306, China; luoyulan0818@163.com (Y.L.); luohua0303@163.com (Z.L.); 3National Demonstration Center for Experimental Fisheries Science Education, Shanghai Ocean University, Shanghai 201306, China

**Keywords:** fishing ban, Yangtze River, fish composition, spatio-temporal pattern, environmental factors, Poyang Lake

## Abstract

Poyang Lake, the largest freshwater lake in China, is one of the two lakes naturally connected to the Yangtze River, playing a vital role in maintaining the ecological balance and biodiversity of the Yangtze River. The government has enforced a fishing ban policy to restore and protect local fish populations. In this study, we assessed the effectiveness of the early stage of the fishing ban by examining and analyzing the fish communities in Poyang Lake and its surrounding waters, and investigating the impacts of environmental factors on fish distribution and growth between 2022 and 2023. These findings not only deepen our understanding of the current status of fish resources in the Poyang Lake basin but also offer a scientific foundation for the restoration and management of the water ecosystem.

## 1. Introduction

The Yangtze River, often referred to as the “Mother River” and the cradle of Chinese civilization, is a global hotspot for biodiversity research. The Yangtze River basin is home to one of the most diverse fish assemblages in the world, with over 4300 aquatic species, including 443 fish species (424 of which are indigenous) [1]. Of these, 194 fish species are endemic to the Yangtze River, and 19 are exotic species [2]. Fish are known as keystone organisms in aquatic ecosystems; their diversity and spatial distribution are crucial indicators of ecological health and significantly influence the ecological functions of the Yangtze River basin [3,4]. However, since the 1960s, the rapid urban economic development and population growth in the basin have driven the expansion of agricultural activities and the unsustainable use of water resources. These changes have caused increased fluctuations in water levels in the basin’s lakes and the gradual loss of wetland areas, posing significant threats to the ecological health and functional integrity of the Yangtze River basin [5,6]. Human activities, including dam construction, industrial pollution, and illegal fencing have disrupted the connectivity between river and lake systems, damaging natural hydrological cycles and exacerbating environmental pressures on both rivers and lakes [7,8,9,10]. These actions have fragmented and reduced fish habitats, greatly impacting fish diversity and altering distribution patterns. Moreover, overfishing has worsened the situation as large numbers of juvenile and spawning fish have been captured, preventing fish populations from recovering and pushing some species to the brink of extinction [11,12,13]. Historical data reveal a sharp decline in the catch of *Coilia nasus*, dropping from 3000 tons in the 1970s to just 4 tons in 2016 [14,15], highlighting a significant reduction in population size. The Biological Integrity Index of the Yangtze River has now reached the critical “fishless” level, signaling a severe depletion of fish resources [16]. A comprehensive survey of the Yangtze River conducted between 2017 and 2021 revealed a notable decline in fish species, including a decrease in indigenous and endemic species, with counts of 323, 293, and 109 species, respectively; in contrast, the number of exotic species rose to 38 [2].

Poyang Lake, situated in the middle reaches of the Yangtze River, is directly connected to the river by a channel that connects it to the Yangtze. As the largest freshwater lake in China and one of the two naturally interconnected lakes in the Yangtze River basin, Poyang Lake plays a vital role in the region’s aquatic ecosystems [17]. Seasonal water level fluctuations and extensive wetland areas provide unique habitats that support a variety of rare aquatic species, playing a significant role in regulating water volumes in the middle and lower reaches of the Yangtze River, while also promoting energy flow and material cycling within the ecosystem. The channel linking Poyang Lake to the Yangtze River is a crucial passage for the exchange of fish populations between the two water bodies. This connection is especially important for many economic freshwater fish species in China (e.g., *Hypophthalmichthys molitrix*, *Ctenopharyngodon idella*, and *Carassius auratus*), which rely on the channel to complete their life cycles [18]. Poyang Lake offers an essential environment for spawning, nurturing, and foraging for these fish, playing an irreplaceable and crucial role in maintaining and improving the diversity and stability of fish populations in the Yangtze River. Similarly, the fishery resources of Poyang Lake have been severely affected by various human-induced stresses. In the 1990s, the average annual catch was 42,600 tons but in the past decade, it has experienced a sharp decline, with a reduction of around 50% [19,20].

Ecosystems around the world are experiencing significant declines in species and populations [21,22,23]. The ongoing decline in fishery resources threatens the ecological balance, disrupts the food chain, and jeopardizes regional biodiversity. To address this pressing issue, the Chinese government established the “Yangtze Office”, assigned long-term monitoring tasks, and allocated special scientific research funding. In 2020, the “Ten-Year Fishing Ban on the Yangtze River” policy was officially enacted, instituting a complete ban on all fishing activities in the river, its main tributaries, lakes, and other crucial fisheries areas. The policy also included promoting artificial fish stocking and release programs, as well as enhancing control over industrial pollution along the river. At the same time, the government placed significant importance on the impact of the fishing ban on fishermen’s livelihoods, providing various support measures to assist their transition to alternative industries, along with implementing economic compensation and social security programs. The goal is to foster the recovery of fishery resources and ecosystems through long-term conservation and restoration. Despite the policy’s success in substantially reducing human interference in Poyang Lake, the lake’s ecosystem and its watershed continue to face significant stress. Our research focused on the key sections of the middle Yangtze River and the inlet area of Poyang Lake, which is crucial for understanding the changes in fish communities during the early stages of the fishing ban, assessing its short-term effects, identifying potential ecological responses, and developing strategies for future ecological restoration.

A significant body of research has examined the biodiversity and distribution of fish in the Poyang Lake basin; however, most of these studies have focused on the main lake area, with limited research on the community structure of fish in both the lake and river systems [24,25,26]. Therefore, based on the implementation of the current fishery closure policy, it is crucial to continuously monitor the community structure and biodiversity of fish species to assess the ecological situation. The primary goal of this study was to characterize the composition and spatial distribution patterns of the fish community during the early stages of the fishing ban and to evaluate the current status of fish resources, providing scientific insights for the effective implementation of future fishing ban policies and the sustainable management of fish resources.

## 2. Materials and Methods

### 2.1. Study Area

Situated in the northern part of Jiangxi Province, China (115°49′~116°46′ E, 28°22′~29°45′ N), Poyang Lake covers a total catchment area of about 160,000 km^2^. This makes it a crucial part of the middle and lower reaches of the Yangtze River. The study area includes the Yangtze River, the channel linking Poyang Lake with the Yangtze River, the northern part of Poyang Lake, Boyang River, Xiuhe River, and its tributary, Liaohe River. A total of five tributaries flow into Poyang Lake, with the southwestern rivers being Boyang River and the Xiuhe River. The study area is vast, including 10 National Aquatic Germplasm Resource Reserves and 2 Provincial Nature Reserves. The topography primarily runs from north to south, gradually transitioning from high mountains to plains, characterized by abundant precipitation with distinct seasonal variations. This has created diverse aquatic environments, including both flowing and static water bodies, each displaying unique ecological characteristics (Figure 1). The river section covering the survey stations (S1~S5) has an average width exceeding 1100 m, with typical flow velocities ranging from 0.06 to 1.6 m/s (>10,000 m^3^/s). In the low hill plain region (S11), the average annual rainfall is about 1560 mm, while in the high hill mountainous region (S15), it exceeds 1850 mm. The river experiences significant seasonal hydrological fluctuations, with notable wetland expansion during the wet season [9,27].

The Poyang Lake basin (PLB) is home to a variety of fish species with different reproductive strategies. Migratory species like *Acipenser sinensis* breed from mid-October to mid-November [28]; on the other hand, most native species such as *C. idella* and *Mylopharyngodon piceus* typically breed between late April and mid-July. The breeding periods of different species are closely linked to the seasonal fluctuations in the Poyang Lake system [29].

### 2.2. Material Sources

The survey was carried out four times during the spring and autumn seasons of 2022–2023 (2022, spring and autumn; 2023, spring and autumn). Eighteen sampling points were set along the main stream of the Yangtze River, the northern area of Poyang Lake, the Boyang River, Xiuhe River and its tributary (Liaohe River) (this survey was part of the Jiangxi Province Aquatic Resources Survey and Monitoring System—Jiujiang Station). Fish sampling used multi-mesh composite gillnets with mesh sizes of 2.0 cm, 6.0 cm, 10.0 cm, and 14.0 cm. Each net was 50 m long, 2 m high, with a total length of 200 m for all the nets combined. In addition, tandem ground cages were employed to collect small fish, with dimensions of 45 cm in width, 33 cm in height, and a mesh size of 0.8 cm (Table 1). For each monitoring section, two sets of multi-mesh gillnets and three sets of tandem ground cages were deployed. In lakes, one set of floating and one set of sinking gillnets were installed. In rivers and other flowing waters, two sets of sinking gillnets were positioned near the shoreline, 10–15 m from the shore. The nets were deployed at 6:00 PM and retrieved at 6:00 AM the following morning, with each fishing duration lasting 12 h. The collected catch was transported to the laboratory for species identification, weighing, and biological measurements. Species were identified at the species level, and the weight was recorded to the nearest 0.1 g. Each sampling point was sampled continuously for five days, providing a relatively consistent fish sample composition. Environmental parameters such as pH, water temperature (Temp), dissolved oxygen (DO), total nitrogen (TN), total phosphorus (TP), and chlorophyll-a (Chl-a) were measured simultaneously using a portable multi-parameter water quality instrument (YSI 6600, Yellow Springs, OH, USA). Water samples were collected on site and sent to the laboratory for the analysis of chemical indicators, including the permanganate index (${\mathrm{MnO}}_{4}^{-}$), ammonia (NH_3_), oil (Oil), and copper (Cu).

### 2.3. Data Analysis

#### 2.3.1. The Index of Relative Importance (IRI)

The relative importance index (*IRI*) [30] was used to measure the degree of ecological dominance of fish communities in each season.(1)$$IRI=\left(N+W\right)\times F\times 10000$$
where *N* is the number of a species as a percentage of the total number of the catch, %; *W* is the biomass of the species as a percentage of the total biomass of the catch, %; and *F* is the number of stations in which the species occurs as a percentage of the total number of stations, %. Species with an *IRI* index of ≥1000 are referred to as dominant species, those with an *IRI* index of 100~1000 are referred to as important species, those with 10~100 are referred to as common species, and those with an *IRI* of <10 are referred to as rare species [31].

#### 2.3.2. Similarity Analysis of Community Structure

The potential spatial clusters of the fish community structure were identified using hierarchical clustering based on the Bray–Curtis similarity matrix. Before computation, fish abundance data were transformed using a log_2_(x+1) transformation [32]. Non-metric multidimensional scaling (NMDS) was used to validate the clustering results and clarify the characteristics of the fish composition in the PLB. The stress coefficient was used to evaluate the performance of the NMDS two-dimensional point array distribution. Generally, a stress value below 0.05 is considered highly representative; values between 0.05 and 0.10 suggest mostly reliable results; while values between 0.10 and 0.20 provide some explanatory value [33].

#### 2.3.3. ANOSIM and SIMPER Analysis

To evaluate the significance of community structure variations among different fish populations, a one-way analysis of similarity (ANOSIM) was applied [32]. Additionally, SIMPER analysis was used to quantify the contribution of specific species to the internal similarity within community structures and the dissimilarities between them [34]. This analysis was based on the species composition and the log_2_(x+1) transformed abundance data of the fish.

#### 2.3.4. Calculation of the Diversity Index

Community species diversity was calculated based on biomass data [35]. The improved formula proposed by Wilhm [36] was employed for the calculation, which is defined as follows:

Shannon–Wiener diversity index (*H*′):(2)$$H\prime =-\sum_{i=1}^{s}\left(P_{i}\mathrm{In}P_{i}\right)$$

Pielou’s evenness index (*J*′):(3)J′=H′/InS 

Margalef’s species richness index (*D*):(4)$$D=\left(S-1\right)/\mathrm{In}N$$
where *S* is the total number of species of fish caught in the study area; *P_i_* is the weight of the *i* th fish species; *N* is the total number of fish weights caught; *J*′ is the homogeneity index; and In*S* is the maximum value of the diversity index.

#### 2.3.5. ABC Curve Analysis

The k-dominance curve provides a convenient method for assessing environmental pollution [37]. Specifically, the k-dominance curve for fish abundance and biomass (Abundance/Biomass Comparison, ABC) was generated using Primer 5.0 software. This method analyzes the characteristics of fish communities under various disturbance conditions by calculating the distribution of species abundance and biomass [37]. When dealing with many stations, sampling times, or replicate samples, plotting ABC curves for each sample can become cumbersome. To streamline the process and assist in statistical analysis, Clarke [38] introduced the statistic *W*. The range of *W* spans from −1 to +1. A positive *W* value indicates uniform species abundance with dominance by a single organism in terms of biomass, while a negative *W* value suggests the opposite.

#### 2.3.6. Redundancy Analysis, RDA

Before conducting the redundancy analysis (RDA), detrended correspondence analysis (DCA) was performed to calculate the gradient lengths (LGA) for each axis. If the maximum gradient length among the four axes was less than 3, RDA was employed; if it exceeded 4, canonical correspondence analysis (CCA) was utilized; and if it lay between 3 and 4, either RDA or CCA were applied [39]. The environmental factors considered in this study include the water temperature (Temp), pH, dissolved oxygen (DO), permanganate index (${\mathrm{MnO}}_{4}^{-}$), total nitrogen (TN), total phosphorus (TP), ammonia (NH_3_), oil (Oil), chlorophyll-a concentration (Chl-a), and copper (Cu). To enhance the accuracy of the results and minimize the influence of dominant species as well as the interference from opportunistic species, logarithmic transformation was applied to both the fish species abundance data and the environmental factor measurements. Analysis of the data was performed using Canoco 5.0 software.

## 3. Results

### 3.1. Species Composition and Dominant Species

This study included four sampling campaigns carried out from 2022 to 2023, twice per year in spring and autumn, resulting in the collection of 49,192 fish specimens with a total biomass of 7017 kg. In spring and autumn, 25,734 and 23,458 fish were captured, with corresponding weights of 3978 kg and 3038 kg, respectively. A total of 120 fish species were identified, classified into 10 orders, 21 families, and 70 genera (Appendix A). In spring and autumn, 104 and 108 species were captured, respectively. Cypriniformes dominated with 79 species, comprising 4 families and 49 genera, accounting for 65.83% of the total species. Siluriformes and Perciformes followed with 19 and 13 species, representing 15.83% and 10.84%, respectively. The remaining orders contained fewer species, making up 7.50% of the total fish species (Figure 2).

Based on the ecological dominance analysis (Table 2), the primary dominant species (*IRI* > 1000) in the PLB are *Hemiculter leucisculus*, *Megalobrama skolkovii*, *H. molitrix*, and *Aristichthys nobilis*. A total of 22 important species (*IRI* > 100) were identified, accounting for 91.41% of the total abundance and 86.46% of the total biomass. In spring, the dominant species include *M. skolkovii*, *A. nobilis*, and *H. leucisculus*, while in autumn, the dominance is represented by *M. skolkovii*, *H. molitrix*, and *C. brachygnathus*. Both spring and autumn contained 23 important species (*IRI* > 100). Among these, *H. leucisculus*, *C. auratus*, *Siniperca chuats*, *Pseudobrama simoni*, and *Culter alburnus* are important fishery species within the Yangtze River basin, with significant economic value. Additionally, endangered species such as *Ochetobius elongatus*, *Myxocyprinus asiaticus*, and *A. sinensis* were captured, along with alien species such as *Cirrhinus mrigala*, *Sinibrama macrops*, *Pseudohemiculter dispar*, *Ictalurus punctatus*, and *Cyprinus carpiovar*. Salt-tolerant fish species *Cynoglossus gracilis* and *Hyporhamphus intermedius* were also observed.

According to the ecological niche (Appendix A), riverine fish species dominate the study area, constituting over 50% of the total, with the highest proportion observed in autumn, at 59.26%. Regarding feeding preferences, the captured species are categorized into omnivorous, carnivorous, herbivorous, and filter-feeding groups, with omnivorous species being the most common, comprising over 55% in each season. Regarding water layer habitats, the majority of fish species live in the middle–lower and bottom layers, accounting for more than 80% of the total population.

### 3.2. Community Similarity

Hierarchical clustering analysis (CA), based on the Bray–Curtis similarity matrix of community average connectivity, shows that all surveyed stations can be spatially grouped into three groups (Figure 3a). The first group (PYS) is mainly distributed across Poyang Lake, the channel connecting the Poyang Lake and the Yangtze River, and the Yangtze River area, showing the highest similarity among these stations. The second group (XBMS) is primarily located in the middle reaches of the Boyang River and Xiuhe River, further from the lake area, showing greater variability in similarity. The third group (XBUS) is mainly found in the upstream areas of the rivers, where the elevation is higher and the distance from the lake area is greater, resulting in a distinct ecological environment compared to the lake area. The one-way ANOSIM results further reveal highly significant differences in fish community structure among the groups (R = 0.893, *p* < 0.01). The NMDS analysis shows a stress value of 0.059, falling within the range of 0.05 < stress < 0.1, indicating that the ordination is generally reliable (Figure 3b).

SIMPER analysis of fish communities across the groups reveals significant or highly significant differences among them. The typical species in each group, along with the divergent species between communities, are consistently the dominant species in the study area; for instance, *C. nasus*, *C. brachygnathus*, *Xenocypris davidi*, and *H. leucisculus* (Table 3 and Table 4).

### 3.3. Diversity Indices

Figure 4 shows the changes in diversity indices across the study area from 2022 to 2023. Overall, significant differences in the Shannon–Wiener diversity index (*H*’), species richness index (*D*), and evenness index (*J*’) are observed among the different areas. Specifically, the PYS exhibits higher values, while the XBMS and XBUS display lower values (Appendix A).

The species richness index (*D*) (Figure 4a) ranges from 1.035 to 4.041. Higher species richness is observed in the Yangtze River area, while lower values are recorded in the middle and upper reaches of the Xiuhe River and Boyang River. Another low-value area is found at S8 and S9, with index values ranging from 2.073 to 2.582. This finding aligns with the River Continuum Concept [40], where species richness in river ecosystems increases progressively from upstream to downstream as habitat diversity increases [41]. The evenness index (*J*’) (Figure 4b) ranges from 0.371 to 0.923, with higher values in the Yangtze River area and lower values near S8 (0.578) and S9 (0.535) in Poyang Lake. In spring, the diversity index (*H*’) (Figure 4c) in the PYS shows the highest average value of 2.577, significantly higher than the 2.208 recorded in the XBMS and 2.432 in the XBUS. The spatial distribution of the diversity index reveals clear differences, with high values at S3, S4, and S5 in the Yangtze River area, ranging from 2.818 to 2.847, while the lowest value is observed near S9, at the confluence of Poyang Lake and the Boyang River (1.883). Regarding seasonal variations, the mean values of the three diversity indices across the entire basin (PLB) remain relatively consistent. In contrast, the mean value for the XBMS shows a slight upward trend, while the PYS and XBUS areas exhibit varying degrees of decline.

The one-way ANOVA results indicated that no significant seasonal variations were found in the fish species richness index (*D*), evenness index (*J*’), or diversity index (*H*’) between spring and autumn within the same region.

### 3.4. Community Stability

The ABC curves of fish communities in the PLB show notable seasonal variations and spatial differences. In the XBMS, the abundance curve initially surpassed the biomass curve and started at a higher point; however, the biomass curve eventually overtook the abundance curve, suggesting that this area is mainly characterized by fast-growing, high-abundance, and small-sized fish species while also containing a few larger species within the community structure (Figure 5b,f). The *W* values for both spring and autumn were negative, measuring −0.035 and −0.029, respectively, indicating that the fish communities were in a critical disturbance state, with slight improvement observed in autumn. In spring, the *W* value for the PYS was 0.049 (Figure 5a), which decreased to 0.038 in autumn (Figure 5e), remaining positive. This suggests that while the fish community experienced minor disturbance, it largely maintained stability. In contrast, the W value for the XBUS increased from 0.061 in spring (Figure 5c) to 0.106 in autumn (Figure 5g), with the biomass starting point higher than in spring. This change indicates a significant increase in the biomass of a particular fish species that has come to dominate the community structure. The *W* value for the entire basin (PLB) is positive in both spring (Figure 5d) and autumn (Figure 5h), showing minimal seasonal variation and a relatively stable fish community structure. In summary, the fish community in the XBUS is more robust than in other areas, whereas the XBMS shows a more pronounced impact compared to the other areas.

### 3.5. Correlation of Communities with Environmental Factors

Redundancy analysis (RDA) was employed to investigate the correlations between environmental factors and fish communities within the study area, incorporating environmental factor parameters (Appendix A). In spring, TP (*F* = 6.4, *p* = 0.002) and temperature (*F* = 6.5, *p* = 0.002) exhibited the most substantial impacts on the fish communities. Specifically, the eigenvalues for the first ordination axis (Axis-1) were 0.4459, and for the second ordination axis (Axis-2), they were 0.1557. These axes collectively explained a cumulative rate of 60.16% for species, with correlation coefficients between species and environmental factors of 0.971 and 0.944, respectively, accounting for 80.50% of the total variation in the species data. Notably, *Pelteobagrus vachelli* and *P*. *nitidus* were significantly influenced by TP and TN, whereas *A. nobilis* and *C. mongolicus* were primarily affected by temperature (Figure 6a).

In autumn, the environmental factor that significantly impacted fish populations in the study area was Oil (*F* = 5.0, *p* = 0.002), followed by temperature and pH (*F* = 2.3, *p* = 0.014; *F* = 2.4, *p* = 0.042). The eigenvalues for the first ordination axis (Axis-1) were 0.3274, while those for the second ordination axis (Axis-2) were 0.1870. Together, Axis-1 and Axis-2 accounted for a cumulative species variation of 51.45%. The correlation coefficients between species and environmental factors were 0.920 and 0.940, respectively, which explained 75.20% of the total variation in the species data. Our findings show that rheophilic species like *Saurogobio dabryi* and *Hemibarbus maculatus* primarily exhibit a positive correlation with dissolved oxygen (DO) and a negative correlation with temperature. These species are mostly found in upstream areas with higher oxygen concentrations and lower temperature gradients. In contrast, species such as *C. dabryi* and *Parabramis pekinensis* dominate the middle and lower reaches of large rivers. In these regions, oil pollution could be spread by navigation, which significantly impacts the distribution of these species (Figure 6b).

## 4. Discussion

### 4.1. Characteristics of Fish Community Structure at the Early Stage of the Fishing Ban

The structural characteristics of fish communities are crucial for understanding ecosystem functions and biodiversity. According to biodiversity theory [42], a community’s diversity depends on the number of species, the uniformity of their abundances, and the complexity of their ecological niches. Due to its unique geographical location and rich biodiversity, the PLB has garnered significant attention and become a key research area [43]. Historical survey data show that during the periods of 1982–1990 [44] and 1997–2000 [45], 103 and 89 fish species were recorded in the main lake area, respectively; in contrast, Yang et al. [46] reported 85 species in their 2018–2019 survey. In this study, conducted during the fishing ban, 120 fish species were documented, surpassing previous records [44,45,46]. In terms of ecological niches, the community was predominantly composed of omnivorous, lower-layer, and demersal fish, consistent with prior findings [47]. It is important to note that differences in the geographical location of sampling stations, survey area selections, sampling times, and methods can lead to discrepancies in the total number of species and endemic species [48]. Further analysis of the ecological niches reveals that the PYS has the highest number of fish species (75), predominantly riverine types (37, 49.3%) with lacustrine types comprising a smaller proportion (18, 24%). River-sea migratory and river-lake migratory types account for 6.7% and 20%, respectively. The PYS shows a significantly higher proportion of omnivorous fish compared to other regions. This community composition reflects the unique ecological characteristics of the PYS as a transitional zone between river and lake ecosystems.

Analysis of diversity indices reveals higher species diversity in areas of high environmental stress, such as the PYS, compared to other areas. This increase in diversity may be attributed to the adaptive strategies of organisms, which, in response to declining water quality, food resources, and habitat availability, expand their food sources and foraging ranges to sustain growth and reproduction. Although the number of fish species captured increased following the fishing ban, the diversity index did not significantly improve. This suggests that while species richness increased, species evenness did not effectively enhance. The morphological characteristics of the ABC curve reflect fishing pressure and environmental changes within the ecosystem, thereby affecting the body size distribution and biomass structure of fish communities [49]. The results demonstrate that the fish community structure in the PYS and XBMS is subject to mild disturbance, remaining at a critical threshold. This may be due to the reduced disturbances, such as fishing pressure, poultry farming, river sand mining, and sewage discharge, following the “Ten-Year Fishing Ban on the Yangtze River” policy. However, the long-term effects of human activities on fish resources and hydrology still require extended recovery [50,51].

Additionally, apart from seasonal floods and droughts [52], abnormal disturbances from extreme weather conditions will significantly affect the fish community structure [53]. For example, in 2022, Poyang Lake underwent an extreme drought, resulting in a significant reduction in the area of the PLB. This fragmentation has led to the division of water bodies into smaller, isolated regions, which has isolated fish populations, limiting their movement and genetic exchange. Furthermore, the reduction in water surface area and ecological changes have caused a significant decline in key food resources—such as plankton and aquatic plants—which directly affect the food supply for fish, posing a significant threat to their survival and reproduction. Our findings also show that *W* values in the Yangtze River, Poyang Lake, and its tributaries decrease in autumn, along with a reduction in fish biomass and uneven species body size distribution. For example, *M. skolkovii* exhibited a polarized distribution characteristic in the weight advantage group within the same water body (Appendix A). In autumn, some migratory species are compelled to migrate further to their wintering grounds due to factors such as declining water temperatures, reduced food resources, and habitat requirements [54,55]. Poyang Lake serves as a crucial breeding habitat for these species in spring, while seasonal environmental changes in autumn are key drivers of their migratory behavior. As a crucial node along migratory routes, Poyang Lake’s ecological role significantly influences the life cycles of these migratory species.

### 4.2. Composition of Dominant Species and Regional Variations

The interaction between rivers and lakes has a profound impact on the hydrology, water quality, and fish community structure of Poyang Lake [56]. Rivers not only supply additional food sources and favorable conditions for fish communities in the lake but also contribute to notable spatial heterogeneity in riverine fish communities due to the differences in water quality, flow rate, and habitat structure (such as differences between mountainous and plain areas) [57]. This spatial heterogeneity enriches the diversity of fish communities in the lake. The influx of rivers into the lake influences both the food availability and habitat conditions for fish, thereby driving dynamic changes in fish community composition.

Dominant species are those that account for a significant proportion of biomass or abundance within a community, and their changes play a critical role in reflecting shifts in the ecological environment and competitive interactions among species [58]. Before the fishing ban, the dominant species in the PLB were primarily high-value, fast-growing fish such as *M. piceus* and *C. auratus*, influenced by fishing activities [19,45]. According to the survey, the dominant species in the study area were mainly omnivorous fish species from the families Cyprinidae and Bagridae. These species typically exhibit strong adaptive and competitive abilities, allowing them to rapidly reproduce and grow in environments with abundant resources [59,60]. In the PYS area, species such as *M. skolkovii*, *H. molitrix*, and *C. nasus* became the new dominant species. In contrast, the XBMS and XBUS regions were characterized by dominant species such as *H. leucisculus*, *Squalidus argentatus*, *X. davidi*, *S. macrops*, and *Rhodeus ocellatus*. These changes were driven by multiple factors, such as water flow velocity, water quality, the availability of food resources, and interspecific competition [61]. At the same time, the implementation of the fishing ban created a relatively stable habitat for these fish species, allowing them to reproduce and thrive, resulting in a shift in the original species composition (Appendix A).

Based on species size, fish in the PYS region tend to be larger individuals with a higher proportion of biomass, while the XBMS and XBUS regions are dominated by smaller fish species with higher abundance. The study found that *C. brachygnathus* and *C. nasus*, typical dominant species in the PYS region, were only captured in that region, with abundance percentages of 39.11% and 15.26%, respectively, and biomass percentages of 2.64% and 3.58%. Regarding seasonal variation, the dominant species’ composition in the PYS and XBUS regions remained stable. However, the composition in the XBMS region changed significantly in autumn, with *C. mongolicus*, *P. simoni*, and *X. argentea* replacing *S. argentatus* as the new dominant species. Additionally, we identified species with high water quality requirements, such as freshwater *Rhinogobius* spp. and *Sinobdella* spp. but their numbers were low, and their body sizes were small. Overall, despite the predominance of Cypriniformes fish, the fish community consists of smaller and less economically valuable species. The composition of dominant species exhibited pronounced spatial heterogeneity. Current studies have shown that factors such as nutrient inputs from rivers and interspecific competition play a significant role in shaping species composition and distribution [62]. Future research should further explore the interactions between these factors and their specific effects on species distribution, with the goal of achieving a more comprehensive understanding of the mechanisms behind spatial heterogeneity in species composition.

### 4.3. Ecological Adaptation of Fish Communities

In riverine environments, particularly within the channel connecting the Poyang Lake and the Yangtze River, the proportion of large fish species captured is notably high. This phenomenon can be explained by the water exchange at the confluence of Poyang Lake and the Yangtze River, which transports abundant nutrients and detritus, thereby promoting nutrient cycling and transport [63]. Therefore, the channel connecting the Poyang Lake and the Yangtze River creates a dynamic ecosystem marked by high biodiversity and a complete food chain [64]. This area is considered a ”buffer zone” for fish, playing a crucial role in their migration and reproduction activities. As a result, the abundance of food sources and optimal habitat conditions attract large fish to these waters, substantially increasing their capture rate. In contrast, smaller and less mobile fish species, such as *S. argentatus*, are more prevalent in the upper reaches of rivers. Seasonally, the average body mass of fish is higher in spring than in autumn (154.61 g/ind > 129.52 g/ind), which is closely linked to the fish spawning season (primarily in spring). After spawning, the gonads undergo regression, which also affects the fish themselves. This change also indicates that fish have made adaptive adjustments to environmental changes. The restoration of hydrological conditions facilitates the smooth migration of anadromous fish to spawning grounds, promoting the growth of fry and replenishing the population in the area. This seasonal variation reflects the ecological needs and growth cycles of fish and demonstrates that fish populations exhibit strong resilience under favorable environmental conditions [65].

Additionally, species may undergo adaptive evolution by expanding their food sources and habitat use in response to environmental changes, leading to significant morphological adaptations [66]. We found that the proportion of streamlined fish species was significantly higher in the PYS compared to other areas, a characteristic particularly important under strong hydrodynamic conditions. Due to the Yangtze River’s large discharge and high velocity [67], fish can obtain more food per unit of time but individuals must maintain stability in turbulent waters to effectively capture prey [68]. Rheophilic fish are better equipped to exploit resources at higher flow velocities [69]. This finding supports the theory of hydrodynamic selection pressures and confirms previous research suggesting that faster-flowing rivers tend to favor streamlined body types in fish [70]. For example, *R. duospilus* and *R. giurinus* prefer hydrostatic environments, while *C. auratus* favors flowing water environments [71,72]. However, it is important to note that this study did not conduct a detailed examination of specific water body morphological conditions, such as flow velocity and depth. Future research should focus on quantifying these hydromorphological conditions to better clarify the relationship between fish habitat selection and the morphological characteristics of water bodies.

Life history directly reflects the reproductive strategies of species and plays a significant role in their adaptation and survival capabilities within specific environments [73]. Our research identified a higher abundance and diversity of migratory fish species, including *Anguilla japonica*, *A. sinensis*, and *A. schrenckii*, compared to previous studies [46]. This finding suggests an improvement in the survival of these migratory fish within the study area’s ecosystem, likely due to better habitat and environmental conditions. Additionally, this finding underscores the critical importance of protecting migratory pathways, spawning grounds, and the integrity of the food web in promoting biodiversity and supporting the coexistence of fish populations.

### 4.4. The Relationship Between Fish Communities and Environmental Factors

Riverine ecological theory provides a key framework for understanding the relationships between fish diversity, communities, and environmental variables [40]. Recent studies have confirmed that fish diversity in the PLB is influenced by various environmental factors, including water temperature, DO, and nutrients [74,75,76]. Additional research has highlighted the impact of various environmental factors on fish communities [77,78]. Using redundancy analysis (RDA), we found that key environmental factors such as TP, temperature, pH, and oil importantly affected the spatial and seasonal distribution of fish communities in the PLB. Certain species showed a strong sensitivity to these environmental changes.

The Yangtze River accounts for around 55% of China’s total inland waterway shipping, making it one of the most important high-grade waterways in the country [79]. However, emissions from various types of ship fuels could have negative effects on aquatic ecosystems. Additionally, improper disposal of waste and domestic garbage generated from navigation activities can lead to water eutrophication and degradation of water quality. Changes in water quality may create new ecological opportunities for certain fish communities to adapt and compete. Our findings reveal that Bagridae and Cyprinidae fish communities in the PYS are significantly correlated with levels of DO, TN, and TP, suggesting that their distribution is mainly influenced by changes in water quality. Notably, most Cyprinidae species have a competitive advantage in eutrophic waters, likely due to their efficient resource utilization and high adaptability to environmental changes [59,80]. These species are better equipped to exploit enriched organic matter, allowing them to maintain higher population densities in areas with high TN/TP ratios. As a result, the diversity and abundance of Cyprinidae species in eutrophic waters are generally higher than those of other fish communities [81].

High-quality habitats provide species with a stable ecological environment that supports reproduction, promotes population growth, and helps maintain ecological balance [82]. Moreover, these habitats are crucial for understanding how species aggregate in specific environments, as they reveal interspecies interactions and provide valuable insights into their life histories [83]. The autumn analysis shows that changes in environmental factors cause certain fish species to respond more strongly to pollutants (such as oil), which may lead to shifts in competitive relationships among species [84]. This dynamic illustrates how sensitive fish communities are to environmental stress, especially as pollution increases or habitats deteriorate, leading to significant changes in species diversity and community structure [85]. This study found a decline in diversity indices during autumn, along with reductions in abundance and biomass compared to spring. This decline may be due to increased environmental stress, which forces sensitive species to migrate adaptively, potentially creating habitat vacancies that impact species richness and diversity. Water quality is the most critical environmental factor influencing the functional structure of PLB fish communities, while habitat quality primarily affects the reproductive and survival conditions of species, thereby impacting the stability of fish populations and ecosystem balance.

The structure of PLB fish communities is greatly influenced by environmental factors, exhibiting significant seasonal and spatial variations. The fishing ban policy has partially aided the recovery of fish resources; however, environmental pollution remains a potential long-term threat to the sustainable use of these resources. Therefore, future fisheries management strategies should include long-term environmental monitoring, with a focus on the dynamic changes in water quality to ensure the sustainability of fish resources and the health of ecosystems.

## 5. Conclusions

This research provides new insights into the dynamics of the fish community within the PLB during the fishing ban and the influence of environmental factors. The fishing ban policy has successfully aided the recovery of fish populations but pollution remains a significant factor threatening the long-term sustainable use of fish resources. The fishing ban has led to a notable increase in the number of fish species within the PLB, with 120 species, and omnivorous fish emerging as the dominant group. The composition of major fish species exhibits notable spatial variability, with the highest species richness found in the channel connecting the Poyang Lake and the Yangtze River. Seasonal analyses indicate that fish have a higher average weight in spring compared to autumn. Environmental factors such as TN and DO play a crucial role in fish distribution, particularly in eutrophic conditions, where cyprinid fish demonstrate a significant competitive advantage.

## Figures and Tables

**Figure 1 animals-15-00433-f001:**
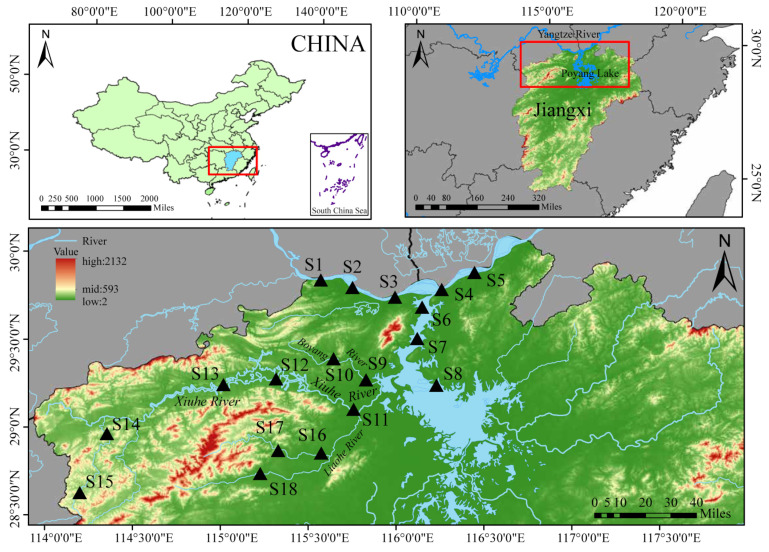
Survey stations for fish sampling in the Poyang Lake basin (each triangle symbol on the map represents a specific survey station (S1–S18), and the map shows the relative positions of these stations within the basin).

**Figure 2 animals-15-00433-f002:**
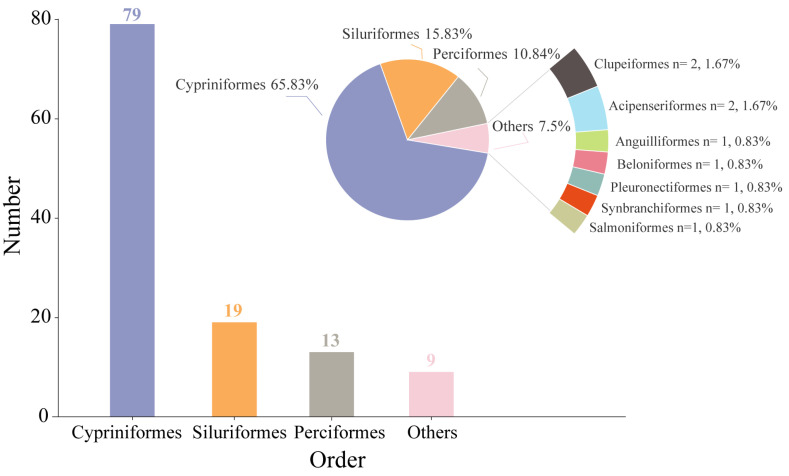
Number and proportion of fish species of different orders in the PLB (the bar chart shows the number of species in each order, and the pie chart illustrates the proportion of each order in relation to the total number of species sampled).

**Figure 3 animals-15-00433-f003:**
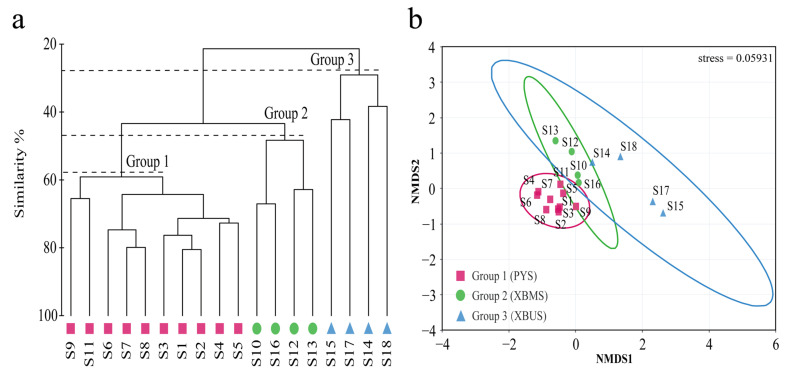
Cluster analysis (**a**) and NMDS (**b**) results of fish communities in the PLB (mapping based on the abundance of species. The magenta square represents the first group (PYS), the green circle represents the second group (XBMS), and the blue triangle represents the third group (XBUS). Same below).

**Figure 4 animals-15-00433-f004:**
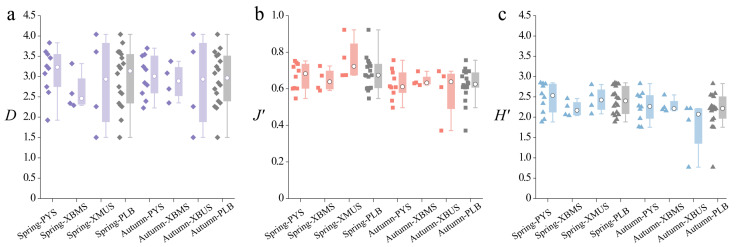
Diversity indices of the fish community in the PLB. (**a**) Species richness index (*D*), (**b**) Evenness index (*J*’), (**c**) Shannon–Wiener diversity index (*H*’) (the prism, square, and circle represent the samples, with the boxplot of each group shown on the right. The white circles represent the median of each group. Colors are used to distinguish between different diversity indices and study areas).

**Figure 5 animals-15-00433-f005:**
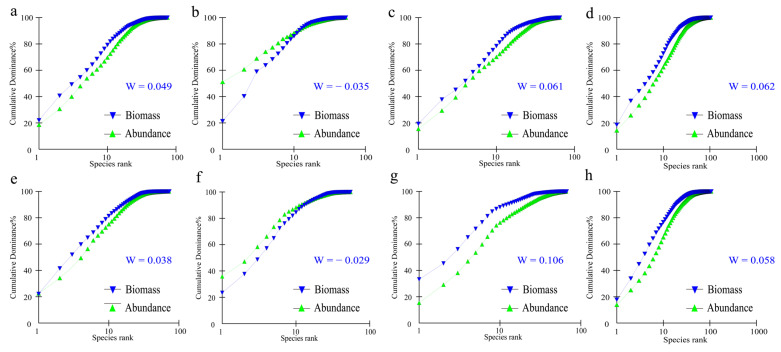
Fish abundance/biomass comparison (ABC) in the PLB ((**a**–**d**) for spring PYS, XBMS, XBUS, and PLB; (**e**–**h**) for autumn PYS, XBMS, XBUS, and PLB. Blue inverted triangles represent biomass, green triangles represent abundance).

**Figure 6 animals-15-00433-f006:**
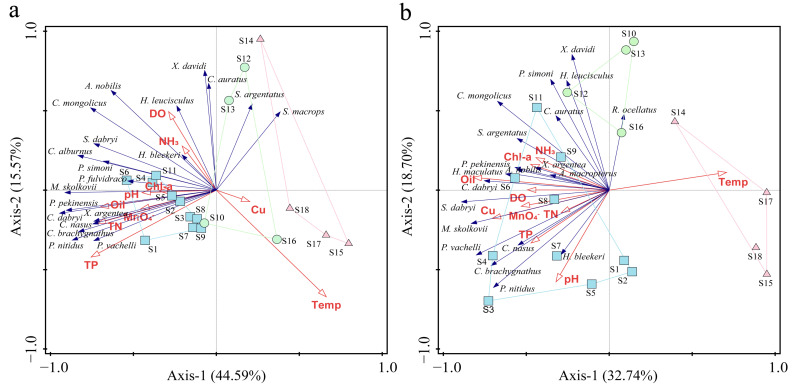
RDA analysis of fish community and environmental factors in spring (**a**) and autumn (**b**) in the PLB (the blue squares, green circles, and pink triangles represent the survey stations included in the PYS, XBMS, and XBUS areas, respectively. The blue arrows represent species, and the red arrows represent environmental factors. Top 20 species in terms of abundance).

**Table 1 animals-15-00433-t001:** Fish sampling surveys and nets details. The survey includes the following parameters: time of survey, downtime, mesh size, type of nets used, and dimensions of the nets (length and height, width and height).

Year	Survey Seasons	Number Stations	Multi-Mesh Gillnets			Tandem Ground Cage			Downtime
			Net Length (m)	Net Height (m)	Mesh Size (cm)	Net Length (m)	Net Width\Height (m)	Mesh Size (cm)	
2022, 2023	Spring, Autumn	18	50	2	2.0	18	0.45\0.33	0.8	12 h
			50	2	6.0				12 h
			50	2	10.0				12 h
			50	2	14.0				12 h

**Table 2 animals-15-00433-t002:** Composition of dominant fish species in the PLB (only species with an *IRI* >100 at any one time in different seasons are listed). Total catch number (*N*, ind), total biomass (*W*, kg).

Species	Spring			Autumn			Total		
	*N*	*W*	*IRI*	*N*	*W*	*IRI*	*N*	*W*	*IRI*
*Megalobrama skolkovii*	1355	738.58	1985.67	1266	540.07	1930.99	2621	1278.66	1962.48
*Aristichthys nobilis*	366	723.07	1415.20	224	331.24	856.33	590	1054.31	1171.73
*Hemiculter leucisculus*	3680	112.62	1237.21	2615	63.51	661.89	6295	176.13	1275.56
*Hypophthalmichthys molitrix*	245	292.93	692.86	233	487.32	1324.73	478	780.26	1007.57
*Coilia brachygnathus*	2947	86.10	756.42	3255	81.41	1011.70	6202	167.50	916.35
*Xenocypris davidi*	762	162.41	508.65	1213	210.23	873.17	1975	372.64	828.91
*Cyprinus carpio*	208	207.85	536.19	204	238.67	727.06	412	446.52	640.06
*Pelteobagrus nitidus*	1945	33.42	653.17	1700	23.91	535.59	3645	57.32	639.85
*Culter mongolicus*	850	133.87	555.63	711	143.66	646.60	1561	277.53	633.63
*Parabramis pekinensis*	560	190.43	580.17	496	136.26	513.26	1056	326.69	566.85
*Culter dabryi*	953	136.20	554.28	438	68.87	298.57	1391	205.08	447.24
*Squalidus argentatus*	1300	17.86	519.49	695	10.62	276.03	1995	28.48	421.35
*Xenocypris argentea*	542	47.14	237.68	1333	57.82	589.98	1875	104.96	412.79
*Coilia nasus*	1478	116.73	433.86	941	91.59	351.29	2419	208.32	394.31
*Carassius auratus*	455	64.34	338.52	475	42.71	343.05	930	107.05	341.61
*Ctenopharyngodon idellus*	248	171.94	381.70	62	43.44	94.10	310	215.38	308.29
*Saurogobio dabryi*	487	10.75	180.23	1104	18.89	444.01	1591	29.65	304.73
*Pseudobrama simoni*	708	19.26	269.61	658	15.52	239.46	1366	34.77	272.70
*Siniperca chuatsi*	124	140.30	267.20	82	61.18	157.55	206	201.49	255.89
*Culter alburnus*	436	71.87	311.15	122	37.83	127.48	558	109.70	239.78
*Pelteobagrus vachelli*	477	47.90	271.78	322	26.99	175.86	799	74.89	239.25
*Pelteobagrus fulvidraco*	569	39.46	320.29	223	13.68	85.60	792	53.14	236.73
*Acheilognathus macropterus*	243	2.36	78.05	841	6.36	295.13	1084	8.72	206.92
*Hemiculter bleekeri*	772	21.17	176.60	1011	3.38	147.36	1783	24.55	198.72
*Culter oxycephaloides*	85	10.76	26.70	160	39.49	121.10	245	50.24	74.19
*Hemibarbus maculatus*	43	7.13	17.32	309	13.75	127.83	352	20.89	73.18

**Table 3 animals-15-00433-t003:** Typical species (contribution rate > 4%) and their contribution to the mean similarity within the group within each sub-basin in the PLB during spring and autumn of 2022–2023.

Species	Spring			Autumn		
	PYS	XBMS	XBUS	PYS	XBMS	XBUS
*Coilia brachygnathus*	6.42			8.80		
*Pelteobagrus nitidus*	6.12			7.23		
*Megalobrama skolkovii*	6.10	4.77		6.83		
*Culter mongolicus*	4.89	4.38		4.26	5.29	
*Culter dabryi*	4.83					
*Coilia nasus*	4.58					
*Parabramis pekinensis*	4.42			4.23		
*Pelteobagrus vachelli*	4.10			4.68		
*Pseudobrama simoni*		10.02			7.38	
*Squalidus argentatus*		9.20	15.67		6.55	
*Saurogobio dabryi*		7.51		7.42	5.90	
*Acheilognathus macropterus*		6.10			6.77	
*Carassius auratus*		5.93	10.60		5.71	19.81
*Xenocypris davidi*		5.78			8.51	
*Hemiculter leucisculus*		5.72			4.14	
*Culter alburnus*		4.29				
*Cyprinus carpio*		4.01			4.19	
*Rhodeus ocellatus*			12.18			
*Opsariichthys bidens*			9.72			4.66
*Pelteobagrus fulvidraco*			5.98		4.19	
*Sinibrama macrops*			5.30			4.10
*Acrossocheilus parallens*			4.89			12.43
*Zacco platypus*			4.16			
*Acrossocheilus kreyenbergii*			4.02			18.80
*Siniperca chuatsi*					4.24	
*Xenocypris argentea*					4.13	
*Pseudorasbora parva*						21.83

**Table 4 animals-15-00433-t004:** Divergent species (contribution rate > 4%) and their contribution to intergroup mean dissimilarity among different sub-basins in the PLB during spring and autumn of 2022–2023.

Species	Spring			Autumn		
	PYS and XBMS	PYS and XBUS	XBMS and XBUS	PYS and XBMS	PYS and XBUS	XBMS and XBUS
*Coilia brachygnathus*	5.77	4.05		5.74	4.81	
*Coilia nasus*	4.72					
*Pseudobrama simoni*			4.15			4.53
*Hemiculter leucisculus*			4.14	4.10		4.15
*Xenocypris davidi*				4.15		4.23
*Pelteobagrus nitidus*				4.85		

## Data Availability

The original contributions presented in this study are included in the article/Appendix A. Further inquiries can be directed to the corresponding authors.

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
