# Peer review of "Post-Fishing Ban Period: The Fish Diversity and Community Structure in the Poyang Lake Basin, Jiangxi Province, China"

_animals, 2025, doi:10.3390/ani15030433_

Round 1

Reviewer 1 Report

Comments and Suggestions for Authors

In the introduction you should provide more information about the “Ten-Year Fishing Ban on the Yangtze River” policies, since it is the main conservation measure, you want to assess.

Line 195. Give more details about the ABC curves. What does the ABC abbreviation mean? Explain how the biomass abundance relationship is connected with environmental conditions

Line 120. As far as I see in the maps and in the text, you included other areas from the Yangtze River (tributaries)  and not only “ the channel connecting the Poyang Lake and the Yangtze River” . You should refer also to these areas in this section.   

Line 121. Which of these five tributaries were included in your research, and why?

Line 122. “The basin is vast….” Which basin you are referring to? Lake's basin? Lake's and the tributaries?

Line 209 - 212 – The transformation in this case was again log2(x+1) as you refer in lines 170 – 171?

Lines 283 – 285 you refer to significant differences between areas but in lines 300 – 303 you mention that no significant differences were observed between the three areas. Please clarify

Figure 3b. Why does the blue circle of the 3rd group also include the stations of the 2nd group? Does this mean that 2nd group is a sub group of the 3rd? Please make it clear. Indicate also in the legend or on the graph the abbreviation of each group (group 1 – PYS, etc)

Lines 310 -313. XBMS area is characterized by small-sized fish or large species that dominate the community structure? 

In the RDA analyses did you consider the inflation factor for the environmental variables?

Lines 371- 374. PYS is located in the lake area and is dominated by riverine type fishes. What happened to the lacustrine species of the lake?

Line 409 – 418. Explain clearly the reasons that may influence the shift in species between the pre and post-ban period and between the lake and the tributaries

Comments on the Quality of English Language

Line 73 – A dot is missing. Rephrase also sentences included in lines 71 to 74 to improve the clarity of their meaning.

Line – 74-75. Rephrase to improve clarity.

Line 78 -82. Complicated sentences, rephrase to improve clarity. You mention in the beginning that you are referring to the connection between the river and the lake. Therefore, there is no need to be repeated afterward.

Line 96-97. This sentence should be at the beginning of your introduction since it’s a more general statement. In this paragraph, you are referring to fishery threats and the Chinese government's actions against them.

Line 341. Refer Culter mongolicus  as C. mongolicus since you refer to the rest of this species in such format  

In the discussion avoid starting too many sentences with the word this (This study, this observation etc).

The quality of the English language needs to be improved. I highly recommend to seek advice from an English-language expert

Author Response

Responses to reviewer 1

Comments 1: In the introduction you should provide more information about the “Ten-Year Fishing Ban on the Yangtze River” policies, since it is the main conservation measure, you want to assess.

Response 1: Thank you very much for your suggestions. We fully agree with your point and, in response to your comments, have included more information about the "Ten-Year Fishing Ban on the Yangtze River" policy in the introduction. Additionally, we have made revisions to the relevant section of the text (Lines 92-99). Once again, we sincerely appreciate your valuable comments.

Comments 2: Line 195. Give more details about the ABC curves. What does the ABC abbreviation mean? Explain how the biomass abundance relationship is connected with environmental conditions.

Response 2: Thank you for your kind comments. We have provided a detailed explanation of the ABC curve and clarified the meaning of the ABC abbreviation (Lines 209-215). The relationship between biomass abundance and environmental conditions is explained as follows [1,2]:

(1) Water quality changes: For instance, pollution in the water may increase the abundance of small, tolerant species, thereby altering the shape of the ABC curve.

(2) Climate change: Variations in temperature or precipitation can affect species distribution and biomass, leading to changes in the curve.

(3) Habitat destruction (especially the impact of fishing activities): Habitat loss may reduce larger species, causing the abundance curve to rise and approach or exceed the biomass curve.

Once again, we sincerely appreciate your comments.

[1] Xu S, Guo J, Liu Y, et al. Evaluation of fish communities in Daya Bay using biomass size spectrum and ABC curve. Frontiers in Environmental Science, 2021, 9, 663169.

[2] Yemane D, Field J G, Leslie R W. Exploring the effects of fishing on fish assemblages using Abundance Biomass Comparison (ABC) curves. ICES Journal of Marine Science, 2005, 62, 374-379.

Comments 3: Line 120. As far as I see in the maps and in the text, you included other areas from the Yangtze River (tributaries) and not only “ the channel connecting the Poyang Lake and the Yangtze River” . You should refer also to these areas in this section.

Response 3: Thank you very much for your suggestions. We have made revisions and additions to the relevant sections of the article (Lines 120-123).

Comments 4: Line 121. Which of these five tributaries were included in your research, and why?

Response 4: Thank you very much for your suggestions. Poyang Lake has five tributaries, among which the Boyang River and the Xiu River, along with their tributaries (Liao River), have been included in our study. The reasons for selecting these rivers are as follows:

(1) Xiu River [3,4]

Geographic features: The main river stretches for 357 km, with several mountains and tributaries along its course, the largest of which is the Liao River. The precipitation distribution varies greatly along the river, with higher precipitation at higher altitudes. The average annual rainfall in the low-mountain plain area (S11) is around 1,560 mm, while in the high-mountain area (S15), it exceeds 1,850 mm.

Hydrological conditions: The river experiences significant seasonal fluctuations in hydrology, with wetland areas expanding greatly during the rainy season. These fluctuations differ markedly across various altitudes, reflecting the impact of topography on hydrological conditions.

Human interference: A total of 618 various reservoirs have been constructed within the basin, significantly affecting local hydrological conditions. Agricultural non-point source pollution and eutrophication are prominent, with considerable impact on water quality.

(2) Boyang River [5]

Geographic features: The main river stretches 105 km, with a limited scale. The terrain and geographical conditions along the river are relatively complex, though the overall size is small.

Hydrological conditions: The average annual water level is 22.46 m, with a maximum level of 29.87 m. The river is highly susceptible to seasonal hydrological fluctuations, with significant water level variations during the rainy and dry seasons. Water quality deteriorates noticeably during the dry season.

Human interference: Due to the limited scale of the river and extensive development of reservoirs, irrigation systems, and hydroelectric projects, the Boyang River is vulnerable to pollution.

The Xiu River and Boyang River, as key tributaries of the Poyang Lake basin, were selected for study due to their unique geographical features, hydrological conditions, and the influence of human activities. These two rivers represent the ecological transition from mountains to plains within the basin and both contain multiple nature reserves, which are vital for maintaining biodiversity and ecosystem health. Furthermore, abundant historical data and monitoring records make it possible to conduct in-depth studies of these rivers. Due to their favorable geographical and hydrological conditions, these rivers are ideal for long-term ecological monitoring and integrated research, enabling a comprehensive assessment of the impact of environmental changes on their aquatic ecosystems.

Once again, we sincerely appreciate your valuable comments.

[3] Committee for the Protection of Urban Construction and Badlands of the Jiujiang People's Congress. Record of the Xiuhe River; Jiangxi People's Publishing House: Nanchang, CN, 2011; pp. 2–86. ISBN 9787210047575.

[4] Li B, Yang G, Wan R, et al. Impacts of hydrological alteration on ecosystem services changes of a large river-connected lake (Poyang Lake), China. Journal of Environmental Management, 2022, 310, 114750.

[5] Sun, X.S. Jiangxi Province: Hydrology,Jiangxi People's Publishing House: Nanchang, CN, 2020; pp. 7–206. ISBN 978-7-210-12693-5.

Comments 5: Line 122. “The basin is vast….” Which basin you are referring to? Lake's basin? Lake's and the tributaries?

Response 5: Thank you again for your suggestions. The basin mentioned in Line 122 refers to the entire study area (the Yangtze River, the channel connecting Poyang Lake and the Yangtze River, the northern area of Poyang Lake, the Boyang River, Xiuhe River, and its tributary). We sincerely apologize for the expression issues in the article and have made the necessary revisions.

Comments 6: Line 209 - 212 – The transformation in this case was again log2(x+1) as you refer in lines 170 – 171?

Response 6: Thank you for your thorough review of our work. The issue you raised regarding the difference in data transformation methods between Lines 170-171 and Lines 209-212 is very important. We would like to provide clarification and explanation. In Lines 170-171, we performed the data analysis using PRIMER 5.0 software, where the transformation method selected was "log2(x+1)". In contrast, in Lines 209-212, we used Canoco 5.0 software, and the transformation method employed was "log". Once again, we sincerely appreciate your valuable comments, which will help improve the quality and transparency of our research.

Comments 7: Lines 283 – 285 you refer to significant differences between areas but in lines 300 – 303 you mention that no significant differences were observed between the three areas. Please clarify.

Response 7: Thank you for your detailed review and valuable comments on our work. The issues you raised regarding the differences in Lines 283-285 and Lines 300-303 are very important, and we would like to clarify and explain them here.

(1) Lines 283-285: This section highlights the significant differences between different regions. Specifically, through analyzing data from various regions, we found that certain indicators exhibit significant differences across regions. These analyses primarily focus on comparisons between different geographical areas.

(2) Lines 300-303: This section indicates that no significant differences were observed between spring and autumn. Specifically, through analyzing data from the same region in different seasons (spring and autumn), we found no significant differences in indicators between these seasons. These analyses are mainly based on seasonal (temporal) changes.

To avoid confusion, we will provide a clearer explanation of these two sections in the article (Lines 315-317). We hope that these revisions will make the article clearer and easier to understand, ensuring that readers can correctly interpret our research findings. Once again, we sincerely appreciate your valuable comments.

Comments 8: Figure 3b. Why does the blue circle of the 3rd group also include the stations of the 2nd group? Does this mean that 2nd group is a sub group of the 3rd? Please make it clear. Indicate also in the legend or on the graph the abbreviation of each group (group 1 – PYS, etc)

Response 8: Thank you for your valuable comments and suggestions on our research. Below is a detailed explanation in response to the issue you raised regarding Figure 3b:

(1) Regarding the overlap of sites between Group 2 and Group 3.

In Figure 3b, the blue circle representing Group 3 includes sites from Group 2, but this does not imply that Group 2 is a subset of Group 3. The overlap may be due to both groups being located in rivers such as the Boyang River and the Xiu River. However, the differences in fish community composition arise from the distinct topographical features (mountainous vs. plain areas) [6]. For example, Group 2 sites are located in the midstream area of the river, with some sites connected to Poyang Lake, where larger fish species forage near the lake mouth [7]. In contrast, Group 3 sites are located upstream, where fish are more likely to be smaller in size. These two groups represent different ecological niches within the river.

(2) Regarding the explanation of abbreviations in the legend.

We greatly appreciate your suggestion. In the next version of the chart, we will clearly label the abbreviations for each group to help readers better understand the classification criteria of the groups (Line 286-288).

Once again, thank you for your careful review and valuable suggestions on our study.

[6] Mondal R, Bhat A. Temporal and environmental drivers of fish-community structure in tropical streams from two contrasting regions in India. PloS one, 2020, 15, e0227354.

[7] Yang G, Zhang Q, Wan R, et al. Lake hydrology, water quality and ecology impacts of altered river–lake interactions: advances in research on the middle Yangtze river. Hydrology Research, 2016, 47, 1-7.

Comments 9: Lines 310 -313. XBMS area is characterized by small-sized fish or large species that dominate the community structure?

Response 9: Thank you for your detailed review of our article and for your attention to the community structure characteristics of the XBMS region. In response to your question, we provide the following clarification.

The XBMS region is primarily dominated by small fish in its community structure. Specifically, in this region, fast-growing, abundant, and small-bodied fish species dominate the community, although there are also a few larger fish species present. To clarify this information, we have revised and strengthened this section (Lines 326-327). Once again, thank you for your valuable comments, which helps improve the clarity and rigor of our research.

Comments 10: In the RDA analyses did you consider the inflation factor for the environmental variables?

Response 10: Thank you for your review and valuable comments on our article. Regarding your question about whether we considered the inflation factor of environmental variables in the RDA analysis: We used Pearson’s correlation coefficient to assess the linear relationships between environmental variables. By calculating the correlation coefficients among all environmental variables, we generated a correlation matrix heatmap to visualize the relationships between the variables (Fig. 1). The results showed that the correlation coefficients between all environmental variables did not exceed 0.8 (|r| ≤ 0.8). Additionally, during the RDA analysis, we applied a log transformation to all environmental variables to improve the accuracy and reliability of the results. Once again, thank you for your valuable comments, which contributes to the rigor and credibility of our research.

Fig 1. Correlation heat map of environmental factors in the Poyang Lake basin in spring (a) and autumn (b).

Comments 11: Lines 371- 374. PYS is located in the lake area and is dominated by riverine type fishes. What happened to the lacustrine species of the lake?

Response 11: Thank you for your thorough review and valuable comments on our research. In response to your question regarding the species variation in the PYS site (Lines 371-374), we provide the following clarification.

The PYS region is located in the Yangtze River (Jiujiang section), the channel connecting Poyang Lake, and the northern Poyang Lake area. This geographical location means that the hydrological conditions of the area are influenced by both the Yangtze River and the connecting channel, characterized by faster water flow and stronger hydrodynamic forces. As a result, the fish community in the PYS region is predominantly composed of riverine species (37 species, 49.3%), significantly higher than the number of lacustrine species (18 species, 24%), diadromous species (5 species, 6.7%), and estuarine-lake migratory species (15 species, 20%). Although PYS includes the northern Poyang Lake area, the number of lacustrine species (18 species, 24%) is lower due to fewer survey sites and the influence of the Yangtze River and the connecting channel. The ecological niche of lacustrine species may be occupied by the more competitive riverine species. Additionally, the presence of estuarine-lake migratory species (15 species, 20%) further supports the ecological characteristics of the PYS region as a river-lake transition zone. These species likely depend on both river and lake habitats, reflecting the pivotal role of the PYS region in the Yangtze and Poyang Lake ecosystems.

We have further refined the relevant analysis and will revise and expand the corresponding sections in the revised manuscript (Lines 388-394). Once again, thank you for your valuable comments.

Comments 12: Line 409 – 418. Explain clearly the reasons that may influence the shift in species between the pre and post-ban period and between the lake and the tributaries.

Response 12: Thank you for your thorough review of our research. Following your suggestions, we have made revisions and additions to the relevant sections in the revised manuscript (Lines 441-454).

Comments 13: Line 73 – A dot is missing. Rephrase also sentences included in lines 71 to 74 to improve the clarity of their meaning.

Response 13: Thank you very much for your comments. We have made revisions to the article (Lines 71-74).

Comments 14: Line – 74-75. Rephrase to improve clarity.

Response 14: Thank you very much for your comments. We have made revisions to the article (Lines 72-74).

Comments 15: Line 78 -82. Complicated sentences, rephrase to improve clarity. You mention in the beginning that you are referring to the connection between the river and the lake. Therefore, there is no need to be repeated afterward.

Response 15: Thank you very much for your comments. We have made revisions to the article (Lines 76-80).

Comments 16: Line 96-97. This sentence should be at the beginning of your introduction since it’s a more general statement. In this paragraph, you are referring to fishery threats and the Chinese government's actions against them.

Response 16: Thank you very much for your comments. We have made revisions to the article (Lines 87-88).

Comments 17: Line 341. Refer Culter mongolicus as C. mongolicus since you refer to the rest of this species in such format.

Response 17: Thank you very much for your comments, and we sincerely apologize for the errors in the article. We have made the necessary revisions to the article.

Comments 18: In the discussion avoid starting too many sentences with the word this (This study, this observation etc).

Response 18: Thank you for your valuable comments and suggestions. In the revised manuscript, we have reduced sentences starting with "This study," "This observation," etc., and have optimized the overall expression.

Comments 19: The quality of the English language needs to be improved. I highly recommend to seek advice from an English-language expert.

Response 19: We greatly appreciate the reviewer’s attention to our research and the valuable suggestions provided. We will seek the assistance of an English language expert to professionally proofread and edit the manuscript. In addition, we will carefully review the entire manuscript to ensure the accurate use of technical terms and ensure the writing is smooth and the logic is clear. Once again, we thank the reviewer for their suggestions. We believe that through this process, our manuscript will be significantly improved and better meet the journal’s publication standards.

Reviewer 2 Report

Comments and Suggestions for Authors

The peer-reviewed manuscript presents the results of an analysis of fish assemblages in the aquatic system of Poyang Lake. The government's fishing moratorium to restore and protect local fish populations is reason enough to conduct research in the area. The results of such a moratorium are relevant to a large local community, but I also see their more general significance. Fish stocks are in crisis in most marine and freshwater areas, so the results of actions such as this will be of interest to many managers and stakeholders. They can help to plan local fish conservation measures in other areas. In this study, the authors evaluated the effectiveness of the early phase of a fishing ban. They examined fish species composition and performed fish assemblage analysis in the Poyang Lake basin. The authors determined the influence of environmental factors on the composition and seasonal distribution of fish assemblages in the water system from 2022 to 2023. The results presented in the manuscript highlight the critical importance of protecting migration routes, spawning grounds and the integrity of the water network to support the biodiversity and coexistence of fish populations. The research topic is interesting and the methodology used is appropriate for this type of analysis. I personally believe that the paper can be accepted for publication if the comments below are followed and some of the issues identified are clarified.

General remarks: 

1. The Poyang Lake basin system is inhabited by many fish species, which are likely to have different reproductive strategies. It will be easier for the reader to assess the seasonal differences in the parameters characterising the fish assemblages if they can find out at what times of the year the fish species found in this water system reproduce. I would like to read about this in the chapter on the study area.

2. The scientific names of the fish need to be checked. I found several inaccuracies in the text. 

3. The nets used to catch the fish are not typical multi-mesh gill nets. The authors used a set of only four mesh sizes with a wide range of sizes, i.e. from 2 to 14 cm. Is this the Chinese standard for multi-mesh gillnets? Why were only these sizes used?

4. The data analysis is comprehensive. The authors used appropriate metrics and data analysis methods. However, I am missing information on what software was used for these analyses. Please provide this.

Specific comments: 

Line 28: Please check the scientific names. For example, Megalobrama skolkovii and Aristichthys nobilis have different scientific names according to FishBase.

Line 31: Hyporamphus intermedius should be Hyporhamphus intermedius.

Lines 40-41: The keywords partly repeat the words in the title. This is an error and should be corrected.

Line 116: What was the depth at the fishing sites? Did it change significantly over the seasons?

Line 129: Can you give the water flow in m3/s?

Lines 143-144: The text needs more details about these traps? Were they fyke nets?

Lines 151-153: How long were the control fish caught in a season? Five days at a sampling site is a long time and there were 18 sites in total. Could different sampling times in different parts of the water system have influenced the results?

Lines 164-167: Is this interpretation of the IRI based on any literature? Or perhaps your own experience? Why were such limits applied?

Lines 206-209: Please explain how the values of the environmental parameters analysed were obtained. Earlier (lines 153-156) it is stated that the environmental parameters were measured using a portable multi-parameter water quality instrument (YSI 6600). Among the parameters mentioned above, there is no information on permanganate index (MnO4-), ammonia (NH3), oil (Oil) or copper (Cu).

Line 208: What do you mean by oil? This term needs clarification. 

Line 213: Why is the chapter called Results and analysis? Results is sufficient. 

Line 217: The precision of the sum of fish biomass is too high. 

Line 226-226: "sum of four species" can be deleted. Such a summation of several species is superfluous. Counting up to three or four is a simple skill.

Line 246: The quality of the figure needs to be improved as the image is not sharp and the small font is difficult to read. 

Line 248: Table 2 - the centred species column does not look good. Perhaps a different text layout would be more reader friendly? E.g. left alignment. The same applies to Tables 3 and 4. 

Lines 286-299: The paragraph needs improvement. As it stands, it is difficult to read the text and follow the data in the figure 4. The referenced values and station numbers do not match what is shown in the figure 4. 

Line 305: The quality of figure 4 is too poor, making the letters blurred and illegible. Figure 'b' does not have a single linear axis. Please consider whether points (prisms, squares and circles) are needed for a box and whisker figure?

Line 328: Again, the quality of the figure needs to be improved. 

Lines 425-427: What could be the reasons for these changes in XBMS?

Comments on the Quality of English Language

Please check the English language for style. Some words may be removed in the places indicated. 

Author Response

Responses to reviewer 2

Comments: The peer-reviewed manuscript presents the results of an analysis of fish assemblages in the aquatic system of Poyang Lake. The government's fishing moratorium to restore and protect local fish populations is reason enough to conduct research in the area. The results of such a moratorium are relevant to a large local community, but I also see their more general significance. Fish stocks are in crisis in most marine and freshwater areas, so the results of actions such as this will be of interest to many managers and stakeholders. They can help to plan local fish conservation measures in other areas. In this study, the authors evaluated the effectiveness of the early phase of a fishing ban. They examined fish species composition and performed fish assemblage analysis in the Poyang Lake basin. The authors determined the influence of environmental factors on the composition and seasonal distribution of fish assemblages in the water system from 2022 to 2023. The results presented in the manuscript highlight the critical importance of protecting migration routes, spawning grounds and the integrity of the water network to support the biodiversity and coexistence of fish populations. The research topic is interesting and the methodology used is appropriate for this type of analysis. I personally believe that the paper can be accepted for publication if the comments below are followed and some of the issues identified are clarified.

Response: Thank you very much for your comments of our research. We are glad to learn that you appreciate the research topic and the methods used. We have carefully read and considered your suggestions and have made the following improvements in the revision.

Comments 1: The Poyang Lake basin system is inhabited by many fish species, which are likely to have different reproductive strategies. It will be easier for the reader to assess the seasonal differences in the parameters characterizing the fish assemblages if they can find out at what times of the year the fish species found in this water system reproduce. I would like to read about this in the chapter on the study area.

Response 1: Thank you for your valuable comments. We have added more details to the description of the study area, particularly regarding fish reproductive strategies (Lines 136-140). Once again, thank you for your valuable suggestions.

Comments 2: The scientific names of the fish need to be checked. I found several inaccuracies in the text.

Response 2: Thank you for your thorough review of our research. We highly value the accuracy of scientific names and have carefully reviewed the scientific names of all fish species in the text. We sincerely apologize for the errors that occurred. After verification, we found that there were indeed some instances of incorrect usage of scientific names, and we have made the necessary corrections.

Comments 3: The nets used to catch the fish are not typical multi-mesh gill nets. The authors used a set of only four mesh sizes with a wide range of sizes, i.e. from 2 to 14 cm. Is this the Chinese standard for multi-mesh gillnets? Why were only these sizes used?

Response 3: Thank you very much for your valuable comments. In the survey of fish resources in the Yangtze River Basin, the Chinese government has set clear requirements for the types and specifications of fishing nets used (triple-layered gillnets, mesh size greater than 2 cm, and net height greater than 1.5 m). We selected four mesh sizes (ranging from 2 cm to 14 cm), based on the Chinese government's requirements, the characteristics of the fishery in the study area, and the fishing needs of target species. The choice of these sizes took several factors into account:

Based on our fisheries practice in the study area, the selection of mesh sizes effectively balances fishing efficiency with the maintenance of species diversity. Different mesh sizes were selected to capture individuals of varying sizes. Smaller mesh sizes are used to catch smaller fish, while larger mesh sizes are suited for capturing larger fish.

In our study, the diversity of fishing gear and the choice of mesh sizes were aimed at more comprehensively capturing fish of different sizes to obtain representative samples of species. This also helps to capture species living in different depths and habitats, ensuring the comprehensiveness and accuracy of the data. Once again, thank you for your valuable comments.

Comments 4: The data analysis is comprehensive. The authors used appropriate metrics and data analysis methods. However, I am missing information on what software was used for these analyses. Please provide this.

Response 4: Thank you for your high praise of the data analysis section of this paper and for pointing out our omission regarding the software used. We apologize for not explicitly listing the software used in the article. We have made revisions to the relevant sections in the manuscript (Lines 209-210, 229).

The data analysis in this paper primarily utilized the following software:

ArcGIS 10.8: Bottom and topographic maps of the study area were downloaded from the official ArcGIS website (Online address: ArcGIS Living Atlas of the World), and the geographic location of the study area was plotted graphically.

Excel: Used for the preliminary organization of fish data and environmental factors, as well as simple calculations, including the calculation of the IRI index.

PRIMER 5.0: Biodiversity, NMDS, ABC curve, and SIMPER analyses were conducted using fish abundance and biomass data.

Canoco 5.0: Correlation analysis between fish community and environmental factors was performed using fish abundance and environmental factor data.

SPSS 26.0: One-way analysis of similarity (ANOSIM) was conducted on the data.

Origin 2021: Graphical representation of the collated and analyzed data was created.

Specific comments:

Comments 5: Line 28: Please check the scientific names. For example, Megalobrama skolkovii and Aristichthys nobilis have different scientific names according to FishBase.

Response 5: Thank you for your suggestion regarding the verification of scientific names. We have carefully checked the scientific names in the text and noticed that there are indeed issues with the scientific names of Megalobrama skolkovii and Aristichthys nobilis. According to FishBase and the latest classification information, the scientific names of M. skolkovii and A. nobilis vary in some literature.

However, many published papers still use the scientific names currently used in our text [1-4]. This discrepancy may be due to differences in classification systems or academic conventions. To maintain academic coherence and align with existing literature, we have decided to continue using these widely accepted scientific names. Once again, thank you for your valuable comments.

[1] Haiqing S, Xiqin H. Effects of dietary animal and plant protein ratios and energy levels on growth and body composition of bream (Megalobrama skolkovii Dybowski) fingerlings. Aquaculture 1994, 127, 189-196.

[2] Chen W, Hubert N, Li Y, et al. Large‐scale DNA barcoding of the subfamily Culterinae (Cypriniformes: Xenocyprididae) in East Asia unveils a geographical scale effect, taxonomic warnings and cryptic diversity. Molecular Ecology, 2022, 31, 3871-3887.

[3] Dong S, Li D. Comparative studies on the feeding selectivity of silver carp Hypophthalmichthys molitrix and bighead carp Aristichthys nobilis. Journal of Fish Biology 1994, 44, 621-626.

[4] Dumitru, G., Ciornea, E., & Vasile, S. The study of some morphological characters at the Aristichthys nobilis species in different stages of development. Lucrări Ştiinţifice, Seria Zootehnie 2011, 55, 16.

Comments 6: Line 31: Hyporamphus intermedius should be Hyporhamphus intermedius.

Response 6: Thank you very much for your suggestion. We sincerely apologize for the error in this section. We have made revisions to the relevant section in the manuscript (Line 31).

Comments 7: Lines 40-41: The keywords partly repeat the words in the title. This is an error and should be corrected.

Response 7: Thank you very much for your suggestion. We have corrected this error.

Comments 8: Line 116: What was the depth at the fishing sites? Did it change significantly over the seasons?

Response 8: Thank you very much for your suggestion. Our survey sites are located 10-15 meters from the shore, and the water depth at the fishing sites ranges from 5 to 15 meters. According to water level data from the Xingzi station (Tab. 1), the water level in Poyang Lake dropped from 18.30m in July 2022 to 6.48m in November 2022, with the dry level being 12.00m. The water depth in the study area shows significant seasonal variation. Once again, thank you for your valuable comments.

Tab. 1 Water level (m) at Xingzi Station of Poyang Lake.

Dates

Jul. 2022

Aug. 2022

Sep. 2022

Oct. 2022

Nov. 2022

Dec. 2022

1

18.30

13.28

8.63

6.85

6.84

7.78

2

18.17

13.07

8.49

6.79

6.83

7.86

3

18.05

12.82

8.39

6.72

6.77

7.97

4

17.89

12.55

8.27

6.79

6.76

8.20

5

17.76

12.24

8.15

6.86

6.77

8.32

6

17.60

11.91

7.93

6.82

6.73

8.37

7

17.59

11.63

7.89

6.88

6.67

8.41

8

17.58

11.37

7.81

7.02

6.70

8.40

9

17.57

11.17

7.74

7.23

6.67

8.35

10

17.49

11.03

7.72

7.49

6.64

8.27

11

17.39

10.86

7.70

7.70

6.64

8.28

12

17.22

10.74

7.68

7.90

6.62

8.11

13

17.04

10.61

7.64

8.05

6.80

7.97

14

16.85

10.52

7.65

8.17

6.74

7.83

15

16.65

10.42

7.58

8.20

6.62

7.68

16

16.40

10.35

7.54

8.15

6.52

7.59

17

16.15

10.29

7.45

8.05

6.49

7.57

18

16.00

10.14

7.39

7.91

6.48

7.36

19

15.83

9.95

7.35

7.71

6.49

7.15

20

15.67

9.78

7.33

7.46

6.49

7.04

21

15.55

9.66

7.33

7.27

6.53

6.98

22

15.44

9.56

7.17

7.14

6.62

6.97

23

15.29

9.49

7.10

7.05

6.69

6.96

24

15.10

9.58

7.12

6.99

6.73

6.94

25

14.90

9.48

7.07

6.92

6.76

6.96

26

14.68

9.32

6.99

6.87

6.80

6.95

27

14.44

9.23

6.97

6.88

6.83

6.93

28

14.21

9.19

6.97

6.88

6.89

6.96

29

13.97

9.03

6.94

6.89

7.16

6.94

30

13.73

8.96

6.88

6.87

7.64

6.92

31

13.51

8.82

6.84

6.86

Max

18.30

13.28

8.63

8.20

7.64

8.41

Min

13.51

8.82

6.88

6.72

6.48

6.86

Avg.

16.26

10.55

7.56

7.27

6.73

7.58

Comments 9: Line 129: Can you give the water flow in m3/s?

Response 9: Thank you very much for your valuable suggestions. We have made revisions to the content in Line 132. The hydrological data is sourced from the Yangtze River Hydrological Network (Online address: www.cjh.com.cn).

Comments 10: Lines 143-144: The text needs more details about these traps? Were they fyke nets?

Response 10: Thank you very much for your suggestion. Fisheries research around the world uses different types of fishing gear based on ecological environments, fishery resources, and target species. For example, in some regions, the choice of fishing gear may prioritize capture efficiency, while in others, it may focus more on ecological protection and sustainable fishing practices. Therefore, it is common for different studies to use various types of fishing gear and mesh sizes, reflecting the specific needs of the study area.

Fisheries management policies and related regulations differ across countries and regions in terms of fishing gear usage. In some countries or regions, governments may set strict mesh size standards to avoid overfishing juvenile fish or to protect the breeding cycles of specific species. In other regions, fishermen and researchers may adjust their choice and use of fishing gear based on actual fishing needs.

In our study, we selected representative fishing gear and mesh sizes based on the research objectives, actual fishing conditions, and policy requirements. The fyke nets you mentioned is referred to as a "tandem ground cages" in China. It serves the same purpose as fyke nets, and we primarily use it to collect benthic fish species.

We aim to provide a detailed description to help readers understand the use of fishing gear and the rationale behind the selection. Once again, thank you for your valuable comments.

Comments 11: Lines 151-153: How long were the control fish caught in a season? Five days at a sampling site is a long time and there were 18 sites in total. Could different sampling times in different parts of the water system have influenced the results?

Response 11: Thank you very much for your suggestion. We place great importance on the potential impact of sampling design on research results. Our specific responses are as follows:

(1) Control group fish capture time: The survey duration followed the standard set by the research design, with a five-day stay at each sampling site. We chose five days as the standard sampling period primarily to ensure the representativeness and stability of the data. This duration effectively captures the target fish species and provides enough time to capture different individuals, avoiding data bias caused by overly frequent sampling.

(2) Sampling cycle: In our seasonal surveys, the sampling cycle is 10-15 days. The duration of each sampling cycle ensures the representativeness of the data and helps capture the diversity and characteristics of the fish community in that season.

(3) Sampling teams and allocation: To ensure consistency and reduce errors, field surveyors were divided into 6 teams (4-6 people per team). Each team was responsible for 2-4 sampling sites and conducted uniform sampling within the same time frame. This arrangement ensures consistent sampling times across different regions, reducing biases caused by time differences in sampling.

(4) Consistency of sampling times across different regions: We ensured consistency in sampling times across different regions by standardizing sampling times and procedures. Each team collected data following the same standards and exchanged information in real-time to ensure the consistency of sampling activities and the comparability of the data.

We believe these measures minimize biases caused by differences in sampling time and location and ensure the scientific validity and reliability of the research results. Once again, thank you for your valuable comments.

Comments 12: Lines 164-167: Is this interpretation of the IRI based on any literature? Or perhaps your own experience? Why were such limits applied?

Response 12: Thank you very much for your suggestion. The interpretation of the IRI in our study is based on existing literature [5]. The classification criteria for the IRI index in our study are based on several classical ecological and fisheries studies [6,7]. We will cite the relevant literature in the text to ensure that the interpretation of the IRI is well-supported by references (Line 179).

[5] Pinkas, L.M.; Oliphant, S.; Iverson, L.K. Food Habits of Albacore, Bluefin Tuna and Bonito in Californian Waters. Fish. Bull. 1971, 152, 1-105.

[6] Xie J, Wang C, Liu L, et al. Assessment of aquatic ecological health based on the characteristics of the fish community structures of the Hun River Basin, Northeastern China. Water, 2023, 15, 501.

[7] Gao J, Liu Z, Jeppesen E. Fish community assemblages changed but biomass remained similar after lake restoration by biomanipulation in a Chinese tropical eutrophic lake. Hydrobiologia, 2014, 724, 127-140.

Comments 13: Lines 206-209: Please explain how the values of the environmental parameters analysed were obtained. Earlier (lines 153-156) it is stated that the environmental parameters were measured using a portable multi-parameter water quality instrument (YSI 6600). Among the parameters mentioned above, there is no information on permanganate index (MnO4-), ammonia (NH3), oil (Oil) or copper (Cu).

Response 13: Thank you very much for your valuable comments. In our study, we used a portable multi-parameter water quality instrument (YSI 6600) to measure various environmental parameters in real-time. The measurement of chemical indicators such as permanganate index (MnO4-), ammonia (NH3), oil (Oil), and copper (Cu) was conducted through laboratory analysis. Specifically, these water samples were collected on-site and sent to the laboratory, where they were tested using standard chemical analysis methods. We have revised and added to the relevant section in the manuscript (Lines 164-166). Once again, thank you for your valuable comments.

Comments 14: Line 208: What do you mean by oil? This term needs clarification. Response 14: Thank you for your valuable comments. The use of the term "oil" could indeed cause some confusion among readers. In this study, the term "oil" refers to the concentration of petroleum-based pollutants in the aquatic environment, such as the introduction of wastewater and exhaust in shipping, along with potential leaks.

Comments 15: Line 213: Why is the chapter called Results and analysis? Results is sufficient.

Response 15: Thank you very much for your suggestion. As per your request, we have made revisions to the relevant section in the manuscript.

Comments 16: Line 217: The precision of the sum of fish biomass is too high.

Response 16: Thank you very much for your suggestion. As per your request, we have made revisions to the relevant section in the manuscript.

Comments 17: Line 226-226: "sum of four species" can be deleted. Such a summation of several species is superfluous. Counting up to three or four is a simple skill.

Response 17: Thank you very much for your suggestion. We have made deletions and revisions to this section.

Comments 18: Line 246: The quality of the figure needs to be improved as the image is not sharp and the small font is difficult to read.

Response 18: Thank you very much for your suggestion. We sincerely apologize for the quality issues with the figures. We will actively work on improving the figures in subsequent versions. Once again, thank you for your valuable comments.

Comments 19: Line 248: Table 2 - the centralized species column does not look good. Perhaps a different text layout would be more reader friendly? E.g. left alignment. The same applies to Tables 3 and 4.

Response 19: Thank you very much for your suggestion. Based on your suggestion, we have optimized the formatting of the tables.

Comments 20: Lines 286-299: The paragraph needs improvement. As it stands, it is difficult to read the text and follow the data in the figure 4. The referenced values and station numbers do not match what is shown in the figure 4.

Response 20: Thank you very much for your suggestion. We have reorganized and optimized the text in Lines 318-319 and cited relevant references to support our results. To improve the clarity and readability of the figures, we have optimized the layout and labels of Figure 4, ensuring that each element (such as the legend and axis labels) is clearer, thus minimizing confusion for readers when interpreting the figure. Once again, thank you for your valuable comments.

Comments 21: Line 305: The quality of figure 4 is too poor, making the letters blurred and illegible. Figure 'b' does not have a single linear axis. Please consider whether points (prisms, squares and circles) are needed for a box and whisker figure?

Response 21: Thank you very much for your suggestion. We sincerely apologize for the poor quality of Figure 4, which caused the letters to be unclear. According to our original version, the resolution of Figure 4 was sufficient, but the image quality may have deteriorated due to the uploading process or system issues. We will ensure that a high-resolution version of Figure 4 is re-uploaded to clearly display all text and details.

We have checked Figure 4's panel 'b' and confirmed that a single-line axis was present in the original version, which may have been caused by a system issue during display or upload. We will ensure that this issue is rechecked and corrected in the final version to ensure the axes are clearly visible.

Regarding the use of dots to represent boxplots and whisker plots, we believe using dots or other symbols in the figure can help differentiate between different indices, thereby improving readability. Once again, thank you for your valuable suggestions.

Comments 22: Line 328: Again, the quality of the figure needs to be improved. Response 22: Thank you very much for your suggestion. We will further optimize the quality of the figures.

Comments 23: Lines 425-427: What could be the reasons for these changes in XBMS?

Response 23: Thank you very much for your suggestion. Changes in flow velocity, water temperature, and the reduction of food resources may directly affect the distribution or growth of species, leading to changes in the composition of dominant species in XBMS. In biological communities within ecosystems (such as fish, plankton, etc.), factors such as species abundance, population structure, or interspecies interactions further drive this change. We have further analyzed the underlying factors of this change and added more details in the manuscript (Lines 447-454, 468-473). Once again, thank you for your valuable suggestions.

Reviewer 3 Report

Comments and Suggestions for Authors

The manuscript contributes new information, concerning the composition of fish communities in the Poyang Lake Basin, China. In parallel, changes are discussed in connection to a recent fishing ban, and some environmental parameters/anthropogenic pressures, as well.

In general, good work was performed: material and methods are enough for safe conclusions. Most of the results are reasonable and well supported by the statistical treatment, which is adequate. Minor methodological issues have to be taken into account, e.g., diversity indices are calculated on biomass, instead of abundance (see attached file). It is not well explained how environmental factors were transformed to feed the statistical analyses, etc.

The interpretation of the results is not the best one: there have been noticed speculations, wrong assumptions, and omissions. The basic biological explanations for proper interpretation are somehow missing, in a sea of facts, results, and even citations. The authors should pay more attention, some statements have to be revised, others skipped, etc. In some cases, the initial interpretation of the obtained results is controversial to known facts, citations, or further discussed matters. It is difficult to accept this text as it is, it should be passed under a more critical scope. In conclusion, the manuscript could be publishable, if these issues are eliminated accordingly. 

Author Response

Responses to reviewer 3

Comments: The manuscript contributes new information, concerning the composition of fish communities in the Poyang Lake Basin, China. In parallel, changes are discussed in connection to a recent fishing ban, and some environmental parameters/anthropogenic pressures, as well.

In general, good work was performed: material and methods are enough for safe conclusions. Most of the results are reasonable and well supported by the statistical treatment, which is adequate. Minor methodological issues have to be taken into account, e.g., diversity indices are calculated on biomass, instead of abundance (see attached file). It is not well explained how environmental factors were transformed to feed the statistical analyses, etc.

The interpretation of the results is not the best one: there have been noticed speculations, wrong assumptions, and omissions. The basic biological explanations for proper interpretation are somehow missing, in a sea of facts, results, and even citations. The authors should pay more attention, some statements have to be revised, others skipped, etc. In some cases, the initial interpretation of the obtained results is controversial to known facts, citations, or further discussed matters. It is difficult to accept this text as it is, it should be passed under a more critical scope. In conclusion, the manuscript could be publishable, if these issues are eliminated accordingly.

Response: First, thank you for your in-depth evaluation and valuable comments on our research. We are pleased that you have recognized the content, methodology, and statistical analysis of our research, and we take your suggestions for improvement very seriously. Below, we will respond to your comments one by one and explain the improvements and corrections made in the revision. We believe these improvements have made the article more rigorous, clearer, and more aligned with the journal's publication standards.

Comments 1: This is confusing. 424 of the 443 are indigenous, or this refers to other species? Please, review the statement/

Response 1: Thank you very much for your suggestion. The original phrasing was ambiguous and could lead readers to believe that the 443 species and 424 species are separate categories. “Recorded 443 fish species (including 424 indigenous species)”. We sincerely apologize for the miscommunication in the text. To eliminate this confusion, we have revised the original text to more clearly convey our research findings (Line 46).

Comments 2: "Fish are known as keystone organisms in aquatic ecosystems, their diversity and spatial distribution are crucial indicators of ecological health and significantly influence the ecological functions of the Yangtze River basin" "

Response 2: Thank you very much for your thorough review and revision suggestions. We fully agree with your revision, which has made the sentence clearer and more fluent, better conveying our research intent. We have made the revision in the manuscript at this section (Lines 47-50). Once again, thank you for your valuable suggestions.

Comments 3: Line 51-55. Citation needed

Response 3: Thank you very much for your careful review of our research. We highly value your suggestion to add references in Lines 51-55. To ensure our argument is well-supported by literature, we have thoroughly reviewed this section and added relevant citations. This will provide more comprehensive support for our argument and ensure the traceability of the information sources. Once again, thank you for your suggestion. We believe these revisions will make the arguments in the article more rigorous and credible.

Comments 4: Line 81. Some, many, rare, protected, other? Please, specify if possible.

Response 4: Thank you very much for your suggestion. We have provided a detailed explanation in the relevant section to enhance the accuracy and readability of the article (Lines 76-80). Once again, thank you for your valuable suggestion.

Comments 5: Line 100. “which” → “which is”

Response 5: Thank you very much for your thorough review of our research and for pointing out the grammatical issue in Line 100. We fully agree with your suggestion and have revised the text by changing "which" to "which is" to ensure the grammatical correctness and fluency of the sentence.

Comments 6: It is better to reformulate as a compact statement, e.g.: "to characterize the composition and spatial distribution patterns of fish communities and to evaluate the current state of fish resources, in this initial period of the fishing ban, in order to furnish scientific insights for the effective enforcement of future fishing ban policies and the sustainable management of fish resources." Or similar text.

Response 6: Thank you very much for your thorough review of our research and for your valuable suggestions. You suggested that we rewrite the "Research Objectives (Lines 110-114)" in a more concise manner to enhance clarity and brevity. We fully agree with this suggestion and have revised the text accordingly (Lines 111-116). Once again, thank you for your suggestion. We believe these improvements will significantly enhance the quality of the article.

Comments 7: Line 120. The lake, the channel and the river, most probably.

Response 7: Thank you very much for your comments. Following your suggestion, we have made the corresponding revision in this section (Lines 120-123).

Comments 8: Line 157. How different sampling data were unified, to be comparable?

Response 8: Thank you very much for your thorough review of our research and for raising the important issue of data comparability. Although two types of fishing gear (multi-mesh gillnets and tandem ground cages) were used in our survey, we ensured sample representativeness and diversity during the sampling process. Multi-mesh gillnets were used to capture fish of various sizes to provide a comprehensive assessment of different fish species, while tandem ground cages were used to capture benthic species, further enriching the sample data of the community structure. The combination of these two types of gear ensured that the samples we collected included fish from different life habits and habitat types, providing a better reflection of the fish community structure and changes in the study area. These sampling methods provided sufficient data support for subsequent statistical analysis.

(1) Unified sampling points: Although two types of fishing gear were used in the survey (For each monitoring section, two sets of multi-mesh gillnets and three sets of tandem ground cages were deployed.), the data collected by both types of gear were treated as data from the same sampling point. During the analysis, all the data were combined and analyzed together, ensuring the integrity of the data.

(2) Standardization: Although two different types of gear were used, we ensured that the sampling time and frequency for all sampling points were consistent to minimize the impact of temporal differences on the data. All collected samples were processed according to the same standards and analysis procedures to ensure consistency in data handling. The raw data were repeatedly checked and verified to ensure the recorded data were accurate and any existing biases were eliminated.

Through these measures, we standardized the sampling data to ensure comparability between sampling points and effectively reflect the overall situation of the sampling sites. Once again, thank you for your valuable suggestion.

Comments 9: Line 162-163. Number of a species' specimens? Please clarify, this is essential. “mass” → “biomass”

Response 9: Thank you very much for your thorough review of our research. You pointed out issues with the "species sample quantity and terminology." We have revised the relevant section of the article (Lines 173-174) and sincerely apologize for the terminology errors.

Comments 10: Line 186. Why biomass, instead of abundance? This could contribute biases-large bodied species are favored in comparison to small ones: a silver carp (30kg) weights more than 100 bleaks (2-3kg). For example, the Shannon-Weiner index (H′) is a commonly used metric in ecology that quantifies the complexity of an ecosystem or community by taking into account both the number of species available and their relative abundance (Nolan and Callahan 2006; Omayio and Mzungu 2019)Please, check it and review accordingly.

Response 10: Thank you very much for your valuable comments. We have carefully considered the issue of calculating the Shannon-Weiner index based on biomass rather than abundance. We understand your concern. However, in the context of our study, calculating the Shannon-Weiner index based on biomass is justified for the following reasons:

Biomass, as a key indicator of energy flow and material cycling in ecosystems, can better reflect species diversity in certain ecosystems, especially when large species contribute significantly to the ecosystem. In the study by Chen et al. [1], relative biomass was used to calculate the diversity index to avoid discrepancies caused by differences in individual sizes [2].

In our study, the ecosystems at the Yangtze River and Poyang Lake monitoring sites are dominated by large species, even though they are less abundant. For example, at site S6, the abundance of the estuarine tapertail anchovy and the bighead carp is 406 and 6, respectively, while the biomass of the estuarine tapertail anchovy (39.44 kg) far exceeds that of the carp (29.76 kg). Similar patterns were observed at sites S4, S7, and S8. Relying solely on abundance, especially when there are large differences in abundance but small differences in biomass, could underestimate the importance of these large species. Furthermore, biomass provides a more accurate reflection of energy distribution among different species in the ecosystem. For instance, a species weighing 20g and another weighing 1500g may have the same influence in a diversity index based on individual numbers, but their actual ecological roles in the ecosystem are different [3]. Using biomass calculation corrects this imbalance. Therefore, we believe that the Shannon-Weiner index based on biomass provides a more comprehensive reflection of species’ ecological roles and contributions, especially in ecosystems dominated by large species.

To provide a more comprehensive view of the results, we calculated the Shannon-Weiner index using both methods. We noticed differences in the Shannon-Weiner index calculated based on abundance and biomass at certain sites and used SPSS 26.0 software for significance analysis. The results showed no significant difference between the Shannon-Weiner index calculated based on biomass and abundance (P > 0.05).

Therefore, we believe that using biomass to calculate the Shannon-Weiner index in this study is reasonable and does not introduce significant bias. We hope this explanation clarifies the reasons for using biomass to calculate the Shannon-Weiner index. We have also revised the relevant section in the manuscript (Lines 198-200). Once again, thank you for your valuable comments.

[1] Chen Y, Shan X, Jin X, et al. Changes in fish diversity and community structure in the central and southern Yellow Sea from 2003 to 2015. Journal of Oceanology and Limnology, 2018, 36, 805-817.

[2] Qiu Y S. The regional changes of fish community on the northern continental shelf of South China Sea. Journal of Fisheries of China, 1988, 12, 303-313.

[3] Wilhm, J.L. Use of biomass units in Shannon's formula. Ecology 1968, 49, 153-156.

Comments 11: Line 206-212. How these parameters were transformed and incorporated into the analysis? Just as raw values, or referent values were also taken into amount? This is crucial, in order to avoid biases on the interpretation of the results.

Response 11: Thank you very much for your thorough review of our research. Specifically, we performed a log transformation of the raw measurements of all environmental factors using Canoco 5.0 software. This transformation helps stabilize variance, thereby improving the reliability of the statistical analysis. During the data processing, we did not directly use reference values for the transformation. Our goal was to use the actual measurements of the observed environmental factors to analyze their relationship with the fish community. However, to ensure comparability of the environmental factor measurements across different sampling sites, we applied standardized sampling and measurement methods for all environmental factors to reduce methodological differences and enhance the reliability of our conclusions. Once again, thank you for your valuable comments. These improvements have helped enhance the rigor and transparency of the article.

Comments 12: Line 215. “surveys” → “sampling campaigns”

Response 12: Thank you very much for your suggestion. We have made revisions to the relevant section in the article (Line 232).

Comments 13: Line 216. “fish” → “fish specimens”; “a combined” → “"total" is better, but there is a repetition of the word in the paragraph”

Response 13: Thank you very much for your suggestion. We have made revisions to the relevant section in the article (Lines 233-234).

Comments 14: Line 226. This is obvious, it can be skipped.

Response 14: Thank you very much for your suggestion. We have made deletions and revisions to this section.

Comments 15: Line 230. This is obvious, it can be skipped.

Response 15: Thank you very much for your suggestion. We have made deletions and revisions to this section.

Comments 16: Line 231. Please, reformulate: what is "both seasons" and "important species"- rare, protected, commercially significant or other?

Response 16: Thank you very much for your thorough review of our research. In this study, "both seasons" specifically refers to spring and summer, and "important species" refers to species with an IRI index greater than 100. To clarify these definitions, we have added the corresponding explanations in the text (Lines 247-248).

Comments 17: Line 234-236. Genus abbreviation is acceptable if it was mentioned before. If the species is mentioned for the first time, the whole name should be written.

Response 17: Thank you very much for your suggestion. In the revised manuscript, we will carefully review the first and subsequent mentions of all species names to ensure that the full genus name is used upon the first mention, and the abbreviation is used in subsequent mentions.

Comments 18: Line 238. “ecological types” → “ecological niches”

Response 18: Thank you very much for your suggestion. We have made revisions to the relevant section in the article (Lines 254).

Comments 19: Line 272. Based on abundance or biomass? Please add in the caption.

Response 19: Thank you very much for your suggestion. We have made revisions to the relevant section in the article (Lines 286-288), specifying the basis for data calculation to ensure the completeness and accuracy of the information.

Comments 20: Line 279. Divergent (fish) species from...area; sampled (when) etc.; an example, how table and picture captions should be more compact and informative. Please, check all of them, and review if necessary.

Response 20: Thank you very much for your thorough review of our research and for your specific suggestions regarding the descriptions of tables and figures. We fully agree with your opinion that the descriptions of tables and figures should be more concise and informative, enabling readers to quickly understand their content and significance. We have reviewed and revised the titles of all tables and figures to ensure they include sufficient key information while avoiding excessive and unnecessary details. This will improve the readability and scientific quality of the article. Once again, thank you for the valuable suggestions from the reviewer.

Comments 21: Line 286-288. This is expectable, species richness increases in parallel to the habitats' diversity, from upstream to downstream. A citation could be added, concerning the river continuum.

Response 21: Thank you very much for your thorough review of our research and for your valuable suggestions regarding the relationship between species richness and habitat diversity. We fully agree with your opinion. To further support this view, we have included a reference to the River Continuum Concept (RCC) in the text. The River Continuum Concept theory suggests that the changes in physical and chemical conditions from the upstream to downstream of a river lead to changes in the biological community structure, thereby influencing species richness [4]. Once again, thank you for your valuable comments, which has been very helpful to our research.

[4] Vannote, R.L.; Minshall, G.W.; Cummins, K.W.; Sedell, J.R.; Cushing, C.E. The river continuum concept. Can. J. Fish. Aquat. Sci. 1980, 37, 130-137.

Comments 22: Line 350. Here, there is a speculation noted. It is obvious, that lowland parts of a riverine ecosystem, accumulate more wastes, than the upstream ones. Fish communities are also normally distributed and altered in zones. Under these circumstances, silver carp e. g., inhabits lower courses of large rivers, where oil could be distributed by navigation. This species is not typical to mountainous streams, where oil splits should be less probable. The same situation is about all other species. Fishes are expected to be mostly sensitive to temperature and oxygen, but which are the gradients for each species? Many rheophilic species live more upstream, where oxygen is more concentrated, and temperature gradient is lower. Under these circumstances, and without additional argumentation, "influenced" is not suitable, and should be changed by "correspond to"

Response 22: Thank you very much for your valuable suggestions. We fully agree with this point. Based on your suggestions, we have provided a more detailed description of the relationship between the distribution of different fish species and environmental conditions, and have adjusted the wording to more accurately reflect these relationships. Additionally, we have included new references to further support our findings. Our aim is to make the article more precise and specific, and to better convey the research findings. Once again, thank you for your valuable comments.

Comments 23: Line 375-377. This is most probably true, but how current data contribute to this conclusion? A speculation again.

Response 23: Thank you very much for your comments. Following your suggestion, we have incorporated diversity index data into our analysis. By comparing with environmental stressors, we have more clearly articulated the relationship between the current data and conclusions to strengthen the argument. Once again, thank you for your valuable suggestions.

Comments 24: Line 378. Repetition, the statement should be reformulated.

Response 24: Thank you very much for your thorough review of our research. Following your suggestion, we have rephrased Line 395-402 to reduce redundancy and improve the clarity and flow of the article.

Comments 25: Line 382. Predation or fishing?

Response 25: Thank you very much for your valuable comments. Both natural predation pressure (such as the impact of fish predators on a particular species) and human fishing activities (such as commercial fishing of fish) can significantly affect fish community structure. In our study, we specifically focus on the impact of human fishing activities on changes in fish community structure. We sincerely apologize for the inappropriate phrasing in the text. Once again, thank you for your valuable suggestions.

Comments 26: Line 384-385. A speculation again, if not supported by exact data. Which results exactly? See previous similar comments. One should normally expect lower species' diversities upstream (more homogeneous habitats), than downstream (more diverse habitats). Moreover, how single and cumulative pressure is estimated on each species? There is no categorization in "sensitive/tolerant species" niches included. The communities' disturbance also? Are there referent conditions to compare? Here, the main survey axis is the fishing ban, but not the environmental variables. How can you estimate their impact, if no older similar data are not included in the analyses?

Response 26: Thank you very much for your suggestion. Our results are based on an analysis of the ABC curve. We acknowledge that this conclusion is based on a preliminary analysis of the existing data and requires support from long-term monitoring data.

In this study, we did not specifically classify "sensitive species" and "tolerant species," but we fully agree with the point you raised. The study primarily analyzed habitat and fish community structure, and this analysis is limited. Therefore, we compared our results with baseline data from relevant studies and literature reports conducted before the fishing ban period [5-7], and analyzed the potential early impacts of the fishing ban on fish community structure using existing data, while speculating on the main factors influencing the fish community. Future studies could adopt standardized ecological stress assessment methods, incorporating aspects such as species growth, reproduction, and habitat adaptability, to further refine the evaluation. We have revised the relevant section in the manuscript and added references (Line 402-409).

Once again, thank you for your valuable comments.

[5] Yin S, Yi Y, Liu Q, et al. A review on effects of human activities on aquatic organisms in the Yangtze River Basin since the 1950s. River 2022, 1, 104-119.

[6] Yang H, Shen L, He Y, et al. Status of aquatic organisms resources and their environments in Yangtze river system (2017–2021). Aquaculture and Fisheries, 2024, 9, 833-850.

[7] Chen T, Wang Y, Gardner C, et al. Threats and protection policies of the aquatic biodiversity in the Yangtze River. Journal for Nature Conservation, 2020, 58: 125931.

Comments 27: Line 388. “lo.ng” → “long”

Response 27: Thank you very much for your suggestion. We have made revisions to the relevant section in the manuscript.

Comments 28: Line 393-394. How this fact could affect current abundance/distribution of all species? At which degree?

Response 28: Thank you very much for your suggestion. According to data from the Xingzi water level station (Tab. 1), the water level in Poyang Lake dropped from 18.30m in July 2022 to 6.48m in November 2022, with the dry level being 12.00m. Due to the long-term effects of the drought, the area of the Poyang Lake basin significantly decreased, and the reduced flow led to the division of the water body into several smaller, isolated regions. The formation of these regions has isolated the spatial distribution of fish populations, limiting their free migration and population exchange. At the same time, the reduction in water area and ecological changes led to a significant decrease in key food resources such as plankton and aquatic plants, directly affecting the food supply for fish, thereby posing a significant threat to their survival and reproduction. The cumulative effect of these factors seriously threatens the ecological balance and sustainability of the fish populations in Poyang Lake. Our research data also indicate that the W value in the study area decreased during this season, along with a reduction in fish biomass and an uneven distribution of species size. We have revised the relevant sections in the manuscript to make the paper more comprehensive and in-depth (Lines 412-422). Once again, thank you for your valuable suggestions.

Tab. 1 Water level (m) at Xingzi Station of Poyang Lake.

Dates

Jul. 2022

Aug. 2022

Sep. 2022

Oct. 2022

Nov. 2022

Dec. 2022

1

18.30

13.28

8.63

6.85

6.84

7.78

2

18.17

13.07

8.49

6.79

6.83

7.86

3

18.05

12.82

8.39

6.72

6.77

7.97

4

17.89

12.55

8.27

6.79

6.76

8.20

5

17.76

12.24

8.15

6.86

6.77

8.32

6

17.60

11.91

7.93

6.82

6.73

8.37

7

17.59

11.63

7.89

6.88

6.67

8.41

8

17.58

11.37

7.81

7.02

6.70

8.40

9

17.57

11.17

7.74

7.23

6.67

8.35

10

17.49

11.03

7.72

7.49

6.64

8.27

11

17.39

10.86

7.70

7.70

6.64

8.28

12

17.22

10.74

7.68

7.90

6.62

8.11

13

17.04

10.61

7.64

8.05

6.80

7.97

14

16.85

10.52

7.65

8.17

6.74

7.83

15

16.65

10.42

7.58

8.20

6.62

7.68

16

16.40

10.35

7.54

8.15

6.52

7.59

17

16.15

10.29

7.45

8.05

6.49

7.57

18

16.00

10.14

7.39

7.91

6.48

7.36

19

15.83

9.95

7.35

7.71

6.49

7.15

20

15.67

9.78

7.33

7.46

6.49

7.04

21

15.55

9.66

7.33

7.27

6.53

6.98

22

15.44

9.56

7.17

7.14

6.62

6.97

23

15.29

9.49

7.10

7.05

6.69

6.96

24

15.10

9.58

7.12

6.99

6.73

6.94

25

14.90

9.48

7.07

6.92

6.76

6.96

26

14.68

9.32

6.99

6.87

6.80

6.95

27

14.44

9.23

6.97

6.88

6.83

6.93

28

14.21

9.19

6.97

6.88

6.89

6.96

29

13.97

9.03

6.94

6.89

7.16

6.94

30

13.73

8.96

6.88

6.87

7.64

6.92

31

13.51

8.82

6.84

6.86

Max

18.30

13.28

8.63

8.20

7.64

8.41

Min

13.51

8.82

6.88

6.72

6.48

6.86

Avg.

16.26

10.55

7.56

7.27

6.73

7.58

Comments 29: Line 395-396. Please, go a step further in the reasonings: Some migratory species are expected to spawn in different habitats during spring, and to return to other areas during autumn, so as to reach their wintering places.

Response 29: Thank you for your valuable suggestions. In response to this issue, we have conducted further in-depth reasoning based on the ecological behavior of migratory species and the results of our study, as detailed below:

In autumn, some migratory species are driven by factors such as decreasing water temperature, reduced food resources, and habitat needs to migrate further to their wintering grounds. Poyang Lake provides an important breeding habitat for these species in the spring, and the environmental changes in autumn further trigger their migratory behavior. As a crucial point in the migratory path, the ecological function of Poyang Lake has a profound impact on the lifecycle of migratory species (Lines 422-428).

Once again, thank you for your valuable suggestions.

Comments 30: Line 430. “communities consists” singular+plural.

Response 30: Thank you very much for your suggestion. We have made corrections to the errors in the text. Once again, thank you for your valuable comments.

Comments 31: Line 431-433. If this is on force only, then the hypothesis concerning the anthropogenic disturbance and nutrients' accumulation on fish communities is standing on the opposite site. In general, the real situation is much more complicated, and cannot be exhaustively described, on the basis of current data only.

Response 31: Thank you very much for your suggestion. We fully agree with this point. We have revised the discussion section: The impact of factors such as nutrient accumulation and interspecies competition on species composition and distribution still requires further investigation. The complex roles of these factors in ecosystems have been confirmed by other studies, but their specific impact in the study area still requires more data and experimental validation. We have made revisions to the relevant section in the manuscript (Lines 468-473). Once again, thank you for your valuable comments.

Comments 32: Line 439. “These the” → “Thus?”

Response 32: Thank you very much for your suggestion. We fully agree with this revision, which has helped improve the clarity and flow of the article. We have made revisions to the relevant section in the manuscript. Once again, thank you for your valuable comments.

Comments 33: Line 440. “create” → “creates”

Response 33: Thank you very much for your suggestion. We have made corrections to the errors in the text. Once again, thank you for your valuable comments.

Comments 34: Line 446-447. It is connected with a decrease of the gonads, due to the spawning period (mainly spring), including the portion spawners.

Response 34: Thank you for your thorough review and valuable suggestions. We fully agree with your view and believe that this perspective is crucial for understanding the research results. This helps to more accurately explain the factors of change observed in the study, enhancing the logical flow and scientific rigor of the article (Lines 486-488). Once again, thank you for the valuable comments from the reviewer. These improvements have helped us enhance the quality of the article.

Comments 35: Line 461. Please, define better: high water levels or other? it is difficult to understand.

Response 35: Thank you for your valuable comments on our research paper. In response to your question about "high hydrodynamic conditions." "High hydrodynamic conditions" refers to the impact of hydrodynamic forces on fish adaptation, not the water level. We realize that this phrasing may not be intuitive for readers. We sincerely apologize for the incorrect description of this definition and have made thorough revisions to ensure clarity and accuracy (Line 499) Once again, thank you for your valuable comments.

Comments 36: Line 464. The exact definition is "rheophilic", not "streamlined"

Response 36: We fully agree with your comments regarding the definition issue. We have reviewed the relevant literature [8,9] and made revisions to the inappropriate sections of the article.

In ichthyology, "rheophilic" refers to fish species that have a preference or adaptation to water flow, typically living in areas with faster currents. These fish possess unique morphological and behavioral traits that enable them to survive and reproduce in turbulent water. In contrast, "streamlined" refers to the body shape of fish being streamlined, which helps reduce water resistance and increase swimming efficiency. Although many rheophilic fish also have streamlined bodies, these two terms are not scientifically equivalent.

In our study, we specifically focus on rheophilic fish, which have a strong dependence on the flow environment, and may exhibit different behaviors and physiological responses in environments with high water levels or significant flow changes.

[8] Lujan, N.K.; Conway, K.W. Life in the fast lane: a review of rheophily in freshwater fishes. Extremophile fishes: Ecology, evolution, and physiology of teleosts in extreme environments, 2015, 107-136.

[9] Meyers, P.J.; Belk, M.C. Shape variation in a benthic stream fish across flow regimes. Hydrobiologia, 2014, 738, 147-154.

Comments 37: Line 465-467. This is not a result, but a widely known fact. Please, add proper citations. In this survey, no hydromorphological conditions (velocity, depth etc.) were studied.

Response 37: Thank you for your thorough review and valuable comments on the paper. We fully agree with this perspective. To ensure academic rigor, we will add appropriate citations in this section to support this statement.

Regarding the study of water body morphological conditions (such as flow velocity, depth, etc.), we acknowledge that these factors were not included in our current survey. We will clearly state this in the discussion section and highlight it as a potential direction for future research. We have made revisions and additions to the relevant sections of the manuscript (Lines 506-510). Once again, thank you for your valuable comments.

Comments 38: Line 470-471. This is not directly illustrated (table, figure other). Moreover, the populations of the first two species spend a lot of time in the sea. On the basis of which assumption the authors conclude population increase in the river-which is just a part of their habitat. From another point of view, there have been performed stockings of Acipenser sinensis. Are stocked fish recognized, or all specimens were assumed as wild?

Response 38: Thank you very much for your valuable comments. Our research conclusions are based on a comparison with previous studies. Specifically, the 2019 survey in Poyang Lake and its tributaries did not record migratory fish species such as Acipenser sinensis and Anguilla japonica, whereas our study, conducted in the Poyang Lake basin in 2022-2023, captured these species, though their abundance was low. This comparison suggests that the Poyang Lake basin may be gradually restoring its ecological functions, with a positive population trend for migratory fish species. Therefore, we did not directly present this comparison in tables or figures in the paper, but we have added the relevant background information to support this discussion (Lines 512-515).

This study speculates that the reappearance of migratory fish species such as Acipenser sinensis and Anguilla japonica in the Poyang Lake basin may be related to recent improvements in the basin’s ecological environment. These improvements include better water quality and habitat restoration in the Poyang Lake basin, providing more suitable living conditions for migratory fish species. Poyang Lake, as an important transitional habitat for migratory fish, may have facilitated the expansion of these species' distribution during the fishing ban period, as its ecological functions were restored [10-12]. These hypotheses are based on the monitoring results of this study combined with previous research findings, supporting the speculation in this paper.

However, as this study only measured the length and weight of the captured Acipenser sinensis, they were subsequently released. We cannot distinguish the specific origin of these individuals (wild or stocked). We expect that future studies, by marking released individuals (e.g., using microchips or genetic markers), can further improve the accuracy of monitoring data. Once again, thank you for your valuable comments.

[10] Wang C Y, Wei Q W, Kynard B, et al. Migrations and movements of adult Chinese sturgeon Acipenser sinensis in the Yangtze River, China. Journal of Fish Biology, 2012, 81(2): 696-713.

[11] Yin H, Wang S, Yang J, et al. Fish community and its relationships with environmental variables in the channel connecting Poyang Lake and the Yangtze River. Aquatic Sciences, 2024, 86, 59.

[12] Jin B, Winemiller K O, Ren W, et al. Basin‐scale approach needed for Yangtze River fisheries restoration. Fish and Fisheries, 2022, 23, 1009-1015.

Comments 39: Line 484. see similar comment above: significally affected is a strong definition, without other similar comparisons for control.

Response 39: Thank you for your review of our research and for your valuable suggestions. In response to your suggestion, we recognize that the phrase "significantly affected" requires a more rigorous definition and supporting comparative analysis. We agree with your point and sincerely apologize for the oversight. In scientific discourse, terminology should be precise and supported by appropriate comparisons and data analysis. To address this issue, we have revised and supplemented the relevant section in the revised manuscript (Line 527). Once again, thank you for your valuable suggestions.

Comments 40: Line 493-496. See similar comments above: are also Bagridae the most tolerant, if yes, provide evidence or citation.

Response 40: Thank you very much for your suggestion. Liu [13] found that the red-tail catfish and longsnout catfish, primarily distributed in the Yangtze River and its tributaries, exhibit strong cold tolerance and are able to adapt to low-temperature environments and survive the winter safely. Additionally, Park [14] reported that fish from the Bagridae family are capable of developing a dual respiratory system through skin breathing to cope with frequent hypoxic conditions in their habitats.

In a recent fish survey in a small river under a highway bridge in Songjiang District, Shanghai, we captured two white-edged bagrid catfish (Pseudobagrus albomarginatus). The habitat conditions were harsh, with turbid water and surface debris such as plastic bags, plastic bottles, and fruit peels floating in the river (Fig. 1). We placed the captured fish in transport boxes and brought them back to the laboratory for further study. After three hours of dry transport, the fish remained alive and showed high vitality when placed back into the water (Fig. 2).

In summary, Bagridae fish exhibit remarkable adaptive traits in cold tolerance, hypoxia tolerance, and resistance to environmental pollution. hese traits are likely the result of long-term adaptation to complex and dynamic aquatic environments and reflect their ecological niche and survival strategies within the ecosystem.

Fig. 1 Pictures of small river habitats in Songjiang District, Shanghai.

Fig. 2 Pseudobagrus brevicorpus (Bagridae).

[13] Liu M, Zhou Y, Guo X, et al. Comparative transcriptomes and metabolomes reveal different tolerance mechanisms to cold stress in two different catfish species. Aquaculture, 2022, 560: 738543.

[14] Park J Y, Kim C H. Habitats and air uptake based on analysis of skin structure of two Korean bullheads, Pseudobagrus brevicorpus and P. koreanus (Pisces; Bagridae). Integrative biosciences, 2007, 11, 155-160.

Round 2

Reviewer 1 Report

Comments and Suggestions for Authors

I believe that the authors have made the necessary corrections. Nevertheless, I would like to suggest a few minor revisions to further improve the presentation of these very important results.

Minor revisions

Comment 1

“The Yangtze River basin is home to the most diverse fish assemblage in the world”

Consider please if it is more appropriate to refer Yangtze River basin as home to one of the most diverse fish assemblages in the world and not the most

Comment 2

There are still a few instances in your text where the genus is either written out in full or only the initial letter is mentioned (e.g. line 79, lines 138-139). I suggest that the first time a species is mentioned in the text, the full name of the genus should be written, and in subsequent mentions, only the initial letter

Comment 3.

In Figure 1. table 1 and line 149 you refer to the sampling as fish resources sampling. Please consider whether it might be more appropriate to refer to these as fish sampling in the text since this section primarily outlines the sampling techniques and locations of fish recordings. I think fish resources as a term is more appropriate to describe the overall availability, diversity, and ecological or economic value of fish populations

Comment 4

In the material and methods section, you mention that “The survey was carried out four times during the spring and autumn seasons of 2022- 2023.

And also in the results “The study included four sampling campaigns carried out from 2022 to 2023, twice each in spring and autumn”

Does this mean that you conducted one survey campaign in Spring of 2022, another one in Autumn of 2022, another one in Spring of 2023, and finally one survey campaign in Autumn of 2023?

However, I see that results and graphs only show differences between spring and summer. Could this mean that there were 2 samplings during the same spring of one year, and 2 samplings in the same autumn of another?

Please clarify the sampling schedules because it is a crucial issue in interpreting your results

Comment 5

Line 311 – 320. Since there is generally sensitivity regarding wild animals that are captured, consider whether it might be better not to refer to the number of specimens but only to abundance and biomass.

Comment 6

In Table 2, a dot is missing, and the legend is in bold letters, presumably by mistake

Comment 7

Lines 383 – 385 “This finding aligns……as habitat diversity increases [41]. Please, consider whether this sentence should be included in the discussion section.

Comment 8

Paragraph lines 380 – 395. It would be quite useful to provide a table in the supplementary materials that lists the values of diversity indices per station and per sampling season. I think this way the spatial differences you are referring to will be clearer. I would also recommend providing a table with the highest lowest, and average values of the environmental variables

Comment 9

In Figure 4. You mention that Prisms, squares, and circles represent each sample. However, I think that rhombus, squares and triangles represent samples.

Comment 10.

Line 469- 473. Is this section intended to present some results of your study? If so, it should perhaps be included in the results, maybe in the Simper section where you present the different species between the grouped areas.

Comment 11.

Line 493. For example is mentioned 2 times

Comment 12.

Lines 494 - 496. Both sentences begin with the phrase “This fragmentation”. Maybe it’s better to merge these two sentences

Comment 13.

Lines 512 "Rivers not supply additional food sources and favorable …" maybe you mean Rivers not only supply additional.

Comment 14

Line 557 the capture proportion or the captured proportion?

Comment 15

Carefully check the references. For example, reference [80] was not found in the text.

Author Response

Responses to reviewer 1 (Round 2)

Comment 1: “The Yangtze River basin is home to the most diverse fish assemblage in the world” Consider please if it is more appropriate to refer Yangtze River basin as home to one of the most diverse fish assemblages in the world and not the most.

Response 1: Thank you very much for your suggestion. We fully understand and agree with your suggestion to use more precise wording. Considering other fish-rich regions around the world (such as the Amazon River basin), describing the Yangtze River basin as "one of the most diverse fish assemblage in the world" would be more accurate and appropriate. We have made the necessary revision to this section in the manuscript (Line 45).

Comment 2: There are still a few instances in your text where the genus is either written out in full or only the initial letter is mentioned (e.g. line 79, lines 138-139). I suggest that the first time a species is mentioned in the text, the full name of the genus should be written, and in subsequent mentions, only the initial letter.

Response 2: Thank you very much for your suggestion. We fully agree with your viewpoint and have reviewed and revised the entire manuscript. Once again, thank you for your valuable comments.

Comment 3: In Figure 1. table 1 and line 149 you refer to the sampling as fish resources sampling. Please consider whether it might be more appropriate to refer to these as fish sampling in the text since this section primarily outlines the sampling techniques and locations of fish recordings. I think fish resources as a term is more appropriate to describe the overall availability, diversity, and ecological or economic value of fish populations.

Response 3: Thank you for your valuable comments. We agree with your point. In this section, the focus is indeed on fish sampling techniques and sampling locations, rather than the availability or ecological and economic value of fish resources. Therefore, changing "fish resource sampling" to "fish sampling" better aligns with the actual content of this section. We have made the changes based on your suggestion to improve the accuracy and clarity of the manuscript. Once again, thank you for your valuable suggestions.

Comment 4: In the material and methods section, you mention that “The survey was carried out four times during the spring and autumn seasons of 2022- 2023.

And also in the results “The study included four sampling campaigns carried out from 2022 to 2023, twice each in spring and autumn”

Does this mean that you conducted one survey campaign in Spring of 2022, another one in Autumn of 2022, another one in Spring of 2023, and finally one survey campaign in Autumn of 2023?

However, I see that results and graphs only show differences between spring and summer. Could this mean that there were 2 samplings during the same spring of one year, and 2 samplings in the same autumn of another? Please clarify the sampling schedules because it is a crucial issue in interpreting your results.

Response 4: Thank you for your detailed review of the sampling times and result descriptions. Your suggestions have been very helpful in clarifying and improving the content of the manuscript. To clarify the sampling times, we confirm that the first survey was conducted in the spring of 2022, the second in the autumn of 2022, the third in the spring of 2023, and the fourth in the autumn of 2023. Thus, the study included one sampling activity each in the spring and autumn of both 2022 and 2023. We have noted that the descriptions in the manuscript were not clear enough, which may have caused some confusion. We have revised the relevant section of the manuscript (Line 148). Once again, thank you for your valuable suggestions.

Comment 5: Line 311 – 320. Since there is generally sensitivity regarding wild animals that are captured, consider whether it might be better not to refer to the number of specimens but only to abundance and biomass.

Response 5: Thank you very much for your attention and suggestions regarding this issue. We understand the sensitivity of referring to captured wildlife, especially regarding potential ethical concerns. However, the species listed in the table are all common species in the basin and are not classified as endangered or protected, so their capture numbers do not raise ethical or legal issues. The main purpose of listing specimen numbers is to calculate the species' IRI (Index of Relative Importance), which is based on both species abundance and biomass, and aims to provide a quantitative assessment of the species' relative importance in the ecosystem. In our study, the specimen numbers provided help to accurately calculate the IRI and conduct scientific analysis. We have ensured that the data provided in the table complies with ethical standards, and the species mentioned do not involve the capture of any protected species. Once again, thank you for your valuable suggestions.

Comment 6: In Table 2, a dot is missing, and the legend is in bold letters, presumably by mistake.

Response 6: Thank you very much for your detailed review of the tables and figures. We have noticed that a dot was indeed missing in Table 2, and the use of bold text in the legend might have caused some confusion. We have addressed both issues based on your suggestions and have ensured that the format of the table and legend is now in accordance with the required standards. Once again, thank you for your valuable comments.

Comment 7: Lines 383 – 385 “This finding aligns……as habitat diversity increases [41]. Please, consider whether this sentence should be included in the discussion section.

Response 7: Thank you very much for your valuable suggestion. We understand your desire for the references to better align with the logical structure of the manuscript. After further consideration, we have decided to retain the reference in the "Results" section, as this section directly supports and explains the relationship between species richness and habitat diversity, which aligns with our observations.

However, we have also strengthened the discussion of the River Continuum Concept in the "Discussion" section, expanding on its role in species richness and habitat diversity. The reference has been appropriately incorporated into this section to support our interpretation of the research findings. We believe that this arrangement makes the manuscript more coherent in structure, while ensuring that the references are applied in the appropriate context. Once again, thank you for your valuable comments.

Comment 8: Paragraph lines 380 – 395. It would be quite useful to provide a table in the supplementary materials that lists the values of diversity indices per station and per sampling season. I think this way the spatial differences you are referring to will be clearer. I would also recommend providing a table with the highest lowest, and average values of the environmental variables.

Response 8: Thank you very much for your suggestion. It has been very helpful for our research. Based on your suggestion, we have added relevant tables for the diversity indices and environmental factors in the attachment. Once again, thank you for your valuable suggestion.

Table S2

Diversity index of the PLB in spring and autumn.

Stations

Spring

Autumn

H

D

J

H

D

J

S1

2.829

3.551

0.743

2.827

3.543

0.756

S2

2.489

3.204

0.666

2.506

3.174

0.689

S3

2.847

3.611

0.752

2.254

3.512

0.607

S4

2.766

3.256

0.736

2.539

3.699

0.656

S5

2.818

3.833

0.720

2.277

3.245

0.613

S6

2.339

3.460

0.604

2.585

2.846

0.711

S7

2.121

2.612

0.601

1.751

2.592

0.496

S8

1.944

2.748

0.547

1.967

2.395

0.578

S9

1.883

1.925

0.601

1.762

2.228

0.535

S10

2.460

3.321

0.672

2.548

3.377

0.696

S11

2.577

3.075

0.699

2.196

2.791

0.608

S12

2.038

2.292

0.605

2.157

2.351

0.634

S13

2.063

2.582

0.590

2.186

2.696

0.620

S14

2.552

3.609

0.675

2.215

3.178

0.609

S15

2.292

1.502

0.923

0.771

1.035

0.371

S16

2.271

2.341

0.724

2.239

3.085

0.630

S17

2.803

4.041

0.771

2.212

2.860

0.696

S18

2.082

2.262

0.674

1.930

2.073

0.668

MAX

2.847

4.041

0.923

2.827

3.699

0.756

MIN

1.883

1.502

0.547

0.771

1.035

0.371

AVG.

2.399

2.957

0.683

2.162

2.816

0.621

Shannon-Wiener diversity index (H'), species richness index (D), and evenness index (J').

Table S3

Spring and autumn environmental factors in the PLB.

Season

Temp

pH

DO

TN

TP

Chl-a

MnO4-

NH3

Oil

Cu

Spring

MAX

28.400

7.895

10.405

2.110

0.110

15.000

4.440

0.014

0.030

0.013

MIN

19.800

6.780

7.050

0.510

0.005

1.000

0.660

0

0

0

AVG.

23.261

7.279

8.677

1.488

0.052

4.639

1.858

0.003

0.014

0.002

Autumn

MAX

26.400

7.630

9.720

4.660

0.190

22.000

3.420

0.007

0.030

0.009

MIN

20.500

6.710

7.050

0.120

0.010

3.000

0.569

0.001

0

0

AVG.

22.861

7.164

8.531

1.399

0.056

9.167

1.703

0.002

0.016

0.004

Water temperature (Temp), pH, dissolved oxygen (DO), total nitrogen (TN), total phosphorus (TP), chlorophyll-a (Chl-a), permanganate index (MnO4⁻), ammonia (NH3), oil (Oil), and copper (Cu).

Comment 9: In Figure 4. You mention that Prisms, squares, and circles represent each sample. However, I think that rhombus, squares and triangles represent samples.

Response 9: Thank you very much for your suggestion. We have made the necessary revisions to the relevant section of the manuscript (Line 330-333).

Comment 10: Line 469- 473. Is this section intended to present some results of your study? If so, it should perhaps be included in the results, maybe in the Simper section where you present the different species between the grouped areas.

Response 10: Thank you for your detailed review and valuable suggestions regarding this section. We understand your suggestion to place this content in the "Results" section or the "SIMPER Analysis" section. However, we believe that this content is more suitable to remain in its current section, as it serves as an extension of the results, aiming to explain the biological significance behind the observed phenomena and data.

This section is not only a direct extension of the results but also plays a crucial role in understanding the reasons for species differences between the different groupings. We hope this section will serve as an interpretive discussion of the analysis results, helping readers better understand the ecological context behind species distribution. Moving it to the "Results" section could disrupt the logical flow of the manuscript, so we have decided to keep it in its current section. Once again, thank you for your valuable suggestions.

Comment 11: Line 493. For example is mentioned 2 times.

Response 11: Thank you very much for your suggestion. We sincerely apologize for the errors that occurred. We have thoroughly reviewed and made the necessary revisions.

Comment 12: Lines 494 - 496. Both sentences begin with the phrase “This fragmentation”. Maybe it’s better to merge these two sentences.

Response 12: Thank you very much for your suggestions regarding language expression. We have noticed that "This fragmentation" was repeated in two sentences. We fully agree with your point and, based on your suggestion, have combined these two sentences into one to improve the conciseness and fluency of the language. We believe this revision makes the sentence more concise and easier to understand. Once again, thank you for your valuable suggestions.

Comment 13: Lines 512 "Rivers not supply additional food sources and favorable …" maybe you mean Rivers not only supply additional.

Response 13: Thank you very much for your suggestion. There was indeed a lack of clarity in that section of the manuscript. We will conduct a thorough review and revision of the manuscript. Once again, thank you for your valuable suggestions.

Comment 14: Line 557 the capture proportion or the captured proportion?

Response 14: Thank you very much for your valuable suggestions. We understand your concern regarding the difference between "capture proportion" and "captured proportion." After careful consideration, we believe that using "capture proportion" is more appropriate in this sentence, for the following reasons:

"Capture proportion" is used to describe the proportion of a particular type of fish (in this case, large fish species) that is captured, based on capture behavior and statistical data. Therefore, "capture" emphasizes the act of catching and its proportion.

"Captured proportion" is typically used to describe the proportion of species that have already been captured in relation to the overall species or population, rather than the proportion of the capture behavior itself. This does not fully align with the meaning we intend to express.

Therefore, using "capture proportion" more accurately conveys the proportion of large fish species during the fishing process.

Comment 15: Carefully check the references. For example, reference [80] was not found in the text.

Response 15: Thank you very much for your suggestion. We have carefully reviewed and revised the references. We sincerely apologize for the errors that occurred. Once again, thank you for your valuable comments.

Reviewer 3 Report

Comments and Suggestions for Authors

In general, most of the issues that were noticed were eliminated.

Figure captions could still be more informative, this issue remains.

After the review, linguistic errors are still present, such as punctuation, missing/improper words, repetitions, etc.

The statement " Cyprinidae species have a competitive advantages in eutrophic waters, likely due to 620 their efficient resource utilization and high adaptability to environmental changes" is not entirely true, despite the citations. In this family, there have been noticed sensitive species, affected even by average anthropogenic disturbance. It is better to state "Most Cyprinidae species"

Author Response

Responses to reviewer 3 (Round 2)

Comment 1: In general, most of the issues that were noticed were eliminated.

Figure captions could still be more informative, this issue remains. After the review, linguistic errors are still present, such as punctuation, missing/improper words, repetitions, etc.

Response 1: Thank you very much for your suggestion. We sincerely apologize for the errors that occurred. We will conduct a thorough review and revision of the manuscript, including adding explanations for the figures to improve the completeness of the information. Once again, thank you for your valuable comments.

Comment 2: The statement "Cyprinidae species have a competitive advantages in eutrophic waters, likely due to their efficient resource utilization and high adaptability to environmental changes" is not entirely true, despite the citations. In this family, there have been noticed sensitive species, affected even by average anthropogenic disturbance. It is better to state "Most Cyprinidae species".

Response 2: Thank you very much for your valuable suggestion. We fully agree with this viewpoint, and based on your suggestion, we have made revisions to the relevant section of the manuscript (Line 555). Once again, thank you for your valuable comments.